# ShorterBetter: Guiding Reasoning Models to Find Optimal Inference Length for Efficient Reasoning

**Jingyang Yi**[*]    **Jiazheng Wang**[*]    **Sida Li**

Data Science Institute, The University of Chicago
Chicago, IL 60637
{jingyang22, jiazhengw, listar2000}@uchicago.edu

## Abstract

Recent models such as OpenAI o1 and DeepSeek-R1 have demonstrated strong performance on reasoning-intensive tasks by generating extended Chain-of-Thought (CoT) traces. While longer reasoning helps with thorough exploration of solution paths for complex problems, it also often leads to inefficient and redundant outputs—a phenomenon commonly described as *overthinking*. In this paper, we propose `ShorterBetter`, a simple yet effective reinforcement learning method that enables reasoning models to learn their own optimal CoT lengths without manual supervision. We define the *Sample Optimal Length* (SOL) as the length of the shortest correct response among multiple generations, which serves as a dynamic reward signal to guide the model toward efficient reasoning. Applied to DeepSeek-Distill-Qwen-1.5B/7B as base models, `ShorterBetter` achieves 50%-80% reduction in output lengths in both in-domain and out-of-domain reasoning tasks while maintaining accuracy. Our reasoning trace analysis shows that `ShorterBetter` refines the structure of the reasoning traces by reducing unnecessary repetition, excessive self-verification, and over-exploration of alternatives.[1]

## 1 Introduction

Improving the reasoning capabilities of Large Language Models (LLMs) has recently led to substantial performance gains in highly structured and complex tasks, such as mathematical problem-solving (Shao et al., 2024) and competitive programming (Jiang et al., 2025). Exemplified by OpenAI o1 (OpenAI, 2024) and DeepSeek-R1 (Guo et al., 2025), these reasoning models are incentivized to generate extended Chain-of-Thought (CoT) steps during inference. Such extended reasoning often includes detailed intermediate steps, reflective self-verifications, and exploration of alternative problem-solving strategies. These advancements are widely believed to enable LLMs to exhibit human-like *System-2 slow thinking* (Evans, 2003; Qu et al., 2025; Li et al., 2025), wherein significant test-time computational resources are deliberately allocated in exchange for improved accuracy and robustness on challenging reasoning benchmarks (Jain et al., 2025; Rein et al., 2024).

However, recent studies have shown that excessively long reasoning processes can be counter-productive, leading not only to computational inefficiencies but also to the so-called *overthinking phenomenon* (Chen et al., 2025; Luo et al., 2025a), where additional reasoning steps yield diminishing returns or even negatively impact model performance (Su et al., 2025).[2] For example, we identify

---

[*]Equal contribution

[1]The source code, model weights, and analysis pipelines are available at: https://jiazheng-wang-yes.github.io/ShorterBetter_ProjectPage/

[2]We provide an experiment in Appendix B.3 demonstrating that incorrect reasoning traces often have longer output lengths.

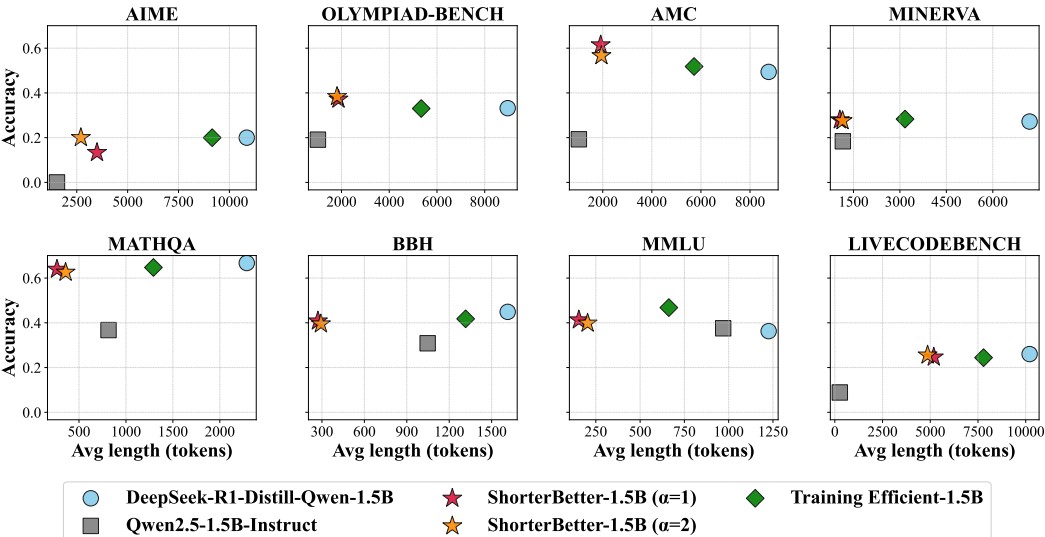

Figure 1: **ShorterBetter exhibits strong length reduction performance (1.5B models).** We report accuracy and output length of five sets of models on various tasks. Models include DeepSeek-R1-Distill-Qwen-1.5B, Qwen2.5-1.5B-Instruct (non-reasoning model), ShorterBetter-1.5B ($\alpha$ =1 and $\alpha$ =2) and Training Efficent-1.5B (Arora and Zanette, 2025). Our method achieves significant length reduction compared to all baseline reasoning models while maintaining accuracy.

several common problematic patterns in lengthy reasoning traces: **(1)** verbose repetitions without new insights, **(2)** early stopping of promising paths, **(3)** continued exploration of alternative solutions despite having arrived at correct answers, and **(4)** collapse of reasoning coherence following very long outputs.[3] These observations indicate that lengthy reasoning traces are highly compressible, motivating techniques producing more concise reasoning chains that improve computational efficiency while preserving overall performance.

Existing approaches targeting efficient LLM reasoning primarily focus on *explicitly controlling reasoning length*, for instance, by imposing a *length budget* based on user specifications (L1; Aggarwal and Welleck (2025)), or by incentivizing shorter reasoning paths through comparisons with reference models (O1-Pruner; Luo et al. (2025a)). Further discussions on related approaches are provided in Section 2. In contrast to these explicit control strategies, our work is based on the hypothesis that reasoning models implicitly possess an *optimal reasoning length* (OL), which is dependent on model capacity and problem difficulty. OL thus represents an in-between "sweet spot": reasoning that is shorter might contain insufficient analysis, while reasoning that is longer risks redundancy and overthinking. This hypothesis has recently undergone theoretical formulation (Wu et al., 2025) and empirical support (Su et al., 2025). Since OL is intrinsic and not directly observable, we approximate it with the *Sample Optimal Length* (SOL): for each problem, we generate multiple outputs and define its SOL as the length of the shortest correct response. Leveraging SOL as a practical proxy for OL, we propose ShorterBetter, a reinforcement learning method that teaches the model to dynamically identify and steer towards efficient reasoning lengths during inference:

In summary, our contributions are as follows.

1. We propose ShorterBetter, a novel reinforcement learning method that guides reasoning model to autonomously learn its optimal reasoning lengths by defining and using Sample Optimal Length as a dynamic reward signal.

2. We empirically demonstrate that ShorterBetter, when applied to 1.5B/7B base models, achieves substantial (50%-80%) reductions in output token length on both in-domain mathematical reasoning tasks and diverse out-of-domain benchmarks, while maintaining accuracy.

3. We introduce novel reasoning trace analysis frameworks to demonstrate how ShorterBetter enhances reasoning efficiency. Specifically, the model reduces verbosity after finding the correct

---

[3]We include concrete examples for each of the aforementioned patterns in Appendix D.

answer and refactors the overall reasoning traces to adopt more confident and targeted reasoning strategies.

## 2 Related Work

**Reasoning in LLMs** Since Wei et al. (2022) demonstrated that CoT prompting enhances LLM reasoning, following research has explored ways to scale test-time computation—such as parallel sampling (Wang et al., 2023) and iterative refinement (Madaan et al., 2023)—to improve performance. This led to Large Reasoning Models (LRMs) that autonomously generate extended CoTs through reinforcement learning-based post-training (Zhang et al., 2025). Examples include OpenAI's o1 (OpenAI, 2024), DeepSeek-R1 (Guo et al., 2025), and Qwen-QwQ (Yang et al., 2024). These models internalize the ability to deliberate over multiple steps before producing final answers, achieving significant performance gains on reasoning-intensive benchmarks (Jimenez et al., 2024; Jain et al., 2025; Gao et al., 2024). Our work addresses a complementary *challenge*: how to dynamically optimize reasoning efficiency without sacrificing accuracy?

**The Overthinking Phenomenon** This challenge, which was initially observed as verbosity and increased computational costs in CoT prompting (Nayab et al., 2024; Han et al., 2024), has become more pronounced with the advent of LRMs—following the identification and understanding of the *overthinking phenomenon* (Chen et al., 2025; Su et al., 2025). This phenomenon is typically understood through two primary lenses. Firstly, overthinking is recognized as a significant source of computational inefficiency. In this view, LRMs expend excessive resources generating overly lengthy and often redundant reasoning traces, especially for simple problems (Lee et al., 2025). This inefficiency manifests as increased token usage, higher latency, and greater computational expenditure (Aytes et al., 2025), and might be exploited adversarially to induce slowdowns (Kumar et al., 2025).

Secondly, overthinking is increasingly understood as a factor that can directly impair practical performance. Fatemi et al. (2025) find that excessive reasoning can degrade logical correctness and introduce irrelevant information; Su et al. (2025) reveal that beyond a certain threshold, longer reasoning does not necessarily improve and can even degrade accuracy, as models may misjudge problem difficulty and introduce compounding errors; similarly, Wu et al. (2025) empirically and theoretically demonstrate a U-shaped performance curve with CoT length, showing that excessively long reasoning paths are susceptible to noise and error accumulation, ultimately reducing correctness. This performance degradation has also been observed in agentic tasks where prolonged internal deliberation correlates with lower task success rates (Cuadron et al., 2025).

**Efficient Reasoning with RL** Many recent and contemporaneous methods have been proposed to control the reasoning length in LRMs. Here, we specifically highlight RL-based post-training strategies directly relevant to our method, leaving readers to the survey by Qu et al. (2025) for an extensive overview. One category of methods uses an explicit *length budget*. For instance, L1 (Aggarwal and Welleck, 2025) treats the budget as a user-specified parameter, while DAST (Shen et al., 2025) adjusts it adaptively based on problem difficulty. These methods then use RL to reward generations that adhere closely to this budget. Other methods design length rewards differently. O1-Pruner (Luo et al., 2025a), for example, compares generated lengths against those from a reference model. More in line with our proposal of sampling multiple responses, Kimi 1.5 (Team, 2025) and Training Efficient (Arora and Zanette, 2025) utilize the shortest and average lengths, respectively, among *all* sampled responses as reward baselines. In contrast, `ShorterBetter` innovatively anchors its reward function to the length of the shortest *correct* response—a design inspired by recent studies on the existence of optimal reasoning lengths in LRMs (Wu et al., 2025; Lee et al., 2025; Su et al., 2025). We include extended comparisons with related and contemporary works in Appendix C.

## 3 Method

In this work, we aim to guide the reasoning model to *think for the right amount of time*, which requires knowing its *optimal reasoning length* (OL)—a prompt-dependent quantity we formally define in Section 3.1. However, calculating OL in practice is challenging due to the absence of precise measures for both problem difficulty and model capability. To address this, we propose in Section 3.2 a sample-based approximation of OL, yielding a practical length reward. Finally, we

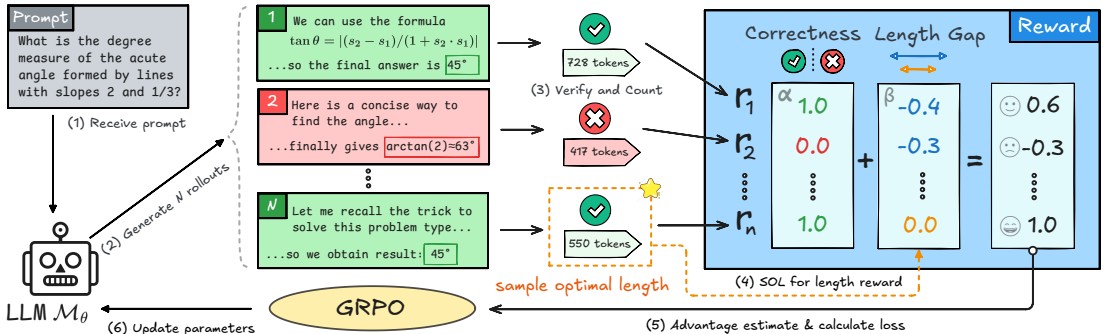

Figure 2: **Flowchart for the `ShorterBetter` pipeline.** We detail the reward design in Section 3.2 and the GRPO algorithm in Section 3.3. Reward values in display are rescaled for better illustration.

integrate this reward design with Group Relative Policy Optimization (GRPO; Shao et al. (2024)) in Section 3.3, resulting in a simple yet effective reinforcement learning method that encourages the model to autonomously discover its optimal reasoning lengths through on-policy learning.

### 3.1 Optimal Reasoning Length (OL): Formal Definition

**Problem Setup** Let $\mathcal{M}_\theta$ be our policy model (i.e. a base LRM) with parameters $\theta$, and $\mathcal{D}$ be the training dataset. Each problem $(x_i, y_i^*) \in \mathcal{D}$ consists of a prompt $x_i$ and a reference response $y_i^*$. Given $x_i$, for each *rollout* the model generates a response $y = (y^1, \ldots, y^t)$ autoregressively from $y \sim p_\theta(\cdot \mid x_i)$, where $y^k$ denotes its $k$-th token and $\ell(y) = t$ is its length. For reasoning tasks, $y$ typically includes long CoT tokens (often enclosed in `</think>` tags) followed by a final answer.

To evaluate response quality, we assume access to a verifier $\mathcal{I}$ that assigns a normalized *correctness score* $s_\mathcal{I}(y, y_i^*) \in [0, 1]$, where higher scores indicate better alignment with $y^*$. For fixed correctness threshold $c \geq 0$ and probability threshold $\epsilon \geq 0$, the optimal reasoning length is defined as

$$\ell_{c,\epsilon}^*(x_i; \theta) := \min\{\ell(y) : y \in \mathcal{Y}_{c,\epsilon}(x_i; \theta)\}, \quad \text{where}$$

$$\mathcal{Y}_{c,\epsilon}(x_i; \theta) := \{y : p_\theta(y \mid x_i) \geq \epsilon \ \wedge \ s_\mathcal{I}(y, y_i^*) \geq c\}.$$

Thus, $\ell_{c,\epsilon}^*(x_i; \theta)$ represents the minimal length at which the model reliably (probability above $\epsilon$) produces a sufficiently correct (score above $c$) response. Intuitively, OL is longer for more difficult problems, as the feasible set $\mathcal{Y}_{c,\epsilon}(x_i; \theta)$ shrinks due to fewer correct responses; conversely, OL becomes shorter when the model $\mathcal{M}_\theta$ is capable of generating concise and correct responses more consistently. This intuition aligns with theoretical insights presented in Wu et al. (2025), despite differences in their specific formulation.

**Remark** The above formulation applies broadly to both closed-ended and open-ended reasoning tasks. For the remainder of this paper, we narrow our focus to tasks with *verifiable answers*, such as mathematical and coding problems. These tasks naturally come with a binary verifier that outputs

$$\mathbb{I}(y = y_i^*) := s_\mathcal{I}(y, y_i^*) \in \{0, 1\},$$

indicating perfect correctness (1) or incorrectness (0). Consequently, we adopt the trivial threshold choice $c = 1$ and use the shorthand notation $\mathbb{I}(y = y_i^*)$ to denote correctness.

### 3.2 Sample Optimal Length (SOL): A Practical Approximation

Directly computing the optimal reasoning length defined in the previous subsection is practically infeasible, as it requires complete knowledge of the model's output distribution. To address this, we introduce the *Sample Optimal Length* (SOL)—a practical, sample-based approximation of OL. Given a prompt $x_i$, our method generates $n$ rollouts of candidate responses $G(x_i) = \{y_1, y_2, \ldots, y_n\}$ from

the policy $p_\theta(\cdot|x_i)$. We then define the SOL for the group $G(x_i)$ as:

$$\ell^{SOL}(G(x_i)) = \begin{cases} \min_{y_j \in G(x_i):\, \mathbb{I}(y_j = y_i^*) = 1} \ell(y_j), & \text{if at least one response is correct,} \\[2mm] \frac{1}{n} \sum_{j=1}^n \ell(y_j), & \text{otherwise.} \end{cases}$$

Given the SOL, we define the following reward function for each response $y_j \in G(x_i)$:

$$r(y_j) = \alpha \cdot \mathbb{I}(y_j = y_i^*) - \beta \cdot \left| \ell(y_j) - \ell^{SOL}(G(x_i)) \right|, \tag{1}$$

where $\alpha, \beta > 0$ are hyperparameters balancing correctness reward and length penalty, respectively.

Our SOL-based reward has several attractive properties:

✓ **No human intervention required**: Unlike explicit length-budget approaches, SOL is entirely data-driven and does not require manual tuning of problem-specific length thresholds.

✓ **Adaptive to difficulty and model capability**: SOL naturally becomes longer for harder problems and shorter for easier ones, as it dynamically adjusts based on the sampled responses; on the other hand, as the model improves during training, SOL automatically updates to reflect the shortest correct reasoning length achievable by the current policy.

✓ **Balanced handling of failure cases**: When no correct responses exist within a sampled group, we set SOL to the average response length as a neutral baseline. Our approach avoids biasing the model toward either overthinking or underthinking when there is no correct signal.

We defer the detailed discussion on specific hyperparameter choices ($\alpha, \beta, n$) to Section 4, and the ablation studies on our reward design to Section 5.3. To see the training dynamics of `ShorterBetter`, please refer to Appendix B.2.

## 3.3 Policy Optimization via GRPO

To optimize the model under the SOL-based reward function, we adopt the Group Relative Policy Optimization (GRPO) algorithm (Shao et al., 2024). GRPO operates on groups of sampled trajectories (in our case, a group $G(x_i)$ has $n$ samples) and assigns within-group *advantage estimates* based on normalizing the rewards $\mathbf{r} = [r(y_1), \cdots, r(y_n)]$ calculated in Eq. (1). Specifically, we have

$$\hat{A}_{j,k} = \frac{r_j - \text{mean}(\mathbf{r})}{\text{std}(\mathbf{r})}$$

as the advantage for the $k$-th step (token) of the $j$-th sample, where 'mean' and 'std' calculate sample reward average and standard deviation. Let $\mathcal{M}_{\theta_\text{ref}}$ be a frozen reference policy (usually a copy of the current model before a few gradient updates), and $p_\theta^k := p_\theta(y_j^k \mid y_j^{<k}, x_i)$ denote the conditional probability on the first $k-1$ tokens, the GRPO loss is

$$\mathcal{L}_{\text{GRPO}}(\theta) = -\frac{1}{n} \sum_{j=1}^n \frac{1}{t_j} \sum_{k=1}^{t_j} \left[ W\left( \hat{A}_{j,k}, p_\theta^k, p_{\theta_{ref}}^k, \varepsilon \right) - \gamma \cdot \tilde{\mathbb{D}}_{\text{KL}} \left[ p_\theta^k \| p_{\theta_{ref}}^k \right] \right], \tag{2}$$

where $W$ denotes a clipped policy improvement objective incorporating the advantage $\hat{A}_{j,k}$ and likelihood ratio between current and reference policies; $\tilde{\mathbb{D}}_{\text{KL}}$ is a KL divergence approximator (Schulman, 2020); $\varepsilon$ and $\gamma$ are hyperparameters for the clipping threshold and KL penalty, respectively. We provide the full expression and details in Appendix A.1.

Policy updates are then performed by minimizing $\mathcal{L}_{\text{GRPO}}(\theta)$ via gradient descent, encouraging the model to increase the likelihood of high-reward samples while constraining deviation from the reference policy. We choose GRPO for its compatibility with group-based reward designs and its empirical success in training LRMs (Guo et al., 2025; Face, 2025). Finally, we include a flowchart illustration of our full method in Fig. 2.

# 4   Experimental Setup

**Datasets**   We fine-tune our base model on DeepScaleR-preview (Luo et al., 2025b), a dataset which comprises 40K mathematical problems drawn from AIME, AMC, Omni-MATH (Gao et al., 2024), and Still dataset (Min et al., 2024). This mixed dataset covers tasks from high-school math problems to Olympiad-level challenges. For in-domain evaluation, we holdout subsets of the training math datasets, containing 1K problems in total. Additionally, we assess the out-of-domain generalization of our method on a set of reasoning-intensive benchmarks with a total of 6K problems, including MathQA (Amini et al., 2019) MMLU (Hendrycks et al., 2021), Big-Bench Hard (BBH; Suzgun et al. (2023)) and coding benchmarks including LiveCodeBench (Jain et al., 2025), MBPP (Austin et al., 2021) and HumanEval (Chen et al., 2021).

**Base Models**   We use DeepSeek-R1-Distill-Qwen-1.5B and 7B (Guo et al., 2025) as our base reasoning models and train our models from them. We will refer to our trained model as `ShorterBetter-1.5B` and `ShorterBetter-7B` in the subsequent sections.

**Evaluations**   For evaluation, we generate one response for each problem. For each benchmark, we consider 3 metrics: (1) **Accuracy**, defined as the number of correct responses divided by the total number of responses; (2) **Output Length**, measured as the average number of tokens per benchmark; and (3) **Accuracy-Efficiency (AE) Score**, a composite metric balancing length reduction against accuracy preservation (introduced by Luo et al. (2025a)). AE Score rewards models that shorten outputs without sacrificing correctness, with higher values indicating better efficiency. For the full formulation and hyperparameter choices, see Appendix A.2.

**Implementation Details**   We used `Verl` (Sheng et al., 2025), an open-source RL training library, for post-training. We fine-tune our 1.5B base model with a batch size of $N = 64$ and a rollout group size of $n = 8$. Training was conducted for 300 steps with a fixed learning rate of $1 \times 10^{-6}$. Since the 7B variant converges faster due to it's better model capacity, we trained for only 200 steps. Due to computational constraints, we set a maximum inference length of 6K tokens for the 1.5B model and 5K tokens for the 7B model during training. (We set 16K maximum tokens budgets during evaluation) These constraints only affect the early stages of training, as the mean inference lengths quickly drop below the limits. To study the effect of hyperparameters, we compared two 1.5B variants with $\alpha = 1$ and $\alpha = 2$, while keeping $\beta = 0.001$ fixed. For the GRPO training configuration, we set $\gamma = 0.001$ and $\varepsilon = 0.2$ in Eq. (2).

**Baselines**   To assess the efficiency of our method, we compare our trained models against 4 baselines:

- **DeepSeek-R1-Distill-Qwen-1.5B/7B** (Guo et al., 2025): Models obtained by distillation from the Qwen series using larger DeepSeek-R1 models. We refer to these models as `Distill-1.5B` and `Distill-7B` in the following sections.

- **Qwen2.5-1.5B/7B-Instruct** (Team, 2024): Instruction-tuned models from the Qwen 2.5 family of large language models. We include these non-reasoning models as a contrast, i.e., fast thinking without deliberate reasoning. We denote them as `Instruct-1.5B` and `Instruct-7B` hereafter.

- **Training Efficient** (Arora and Zanette, 2025): We evaluated the 1.5B/7B open-source models released in this work, which have the same base models as ours.

- **O1-Pruner** (Luo et al., 2025a): We evaluate the 7B open-source model proposed in this work,[4] which is fine-tuned from the Marco-o1 base model (Zhao et al., 2024).

# 5   Results and Discussion

In this section, we present and analyze the effectiveness of our proposed method. We compare the performance of our trained models against the baselines across a range of in-domain math tasks

---

[4]Since this work does not provide a 1.5B model, we will only use 7B version as the baseline.

| | Distill | | Instruct | | ShorterBetter ($\alpha = 2$) | | | Training Efficient | | | O1-pruner | | |
|---|---|---|---|---|---|---|---|---|---|---|---|---|---|
| | Acc. | Len. | Acc. | Len. | Acc. | Len. | AE Score↑ | Acc. | Len. | AE Score↑ | Acc. | Len. | AE Score↑ |
| *In-domain Tasks* | | | | | | | | | | | | | |
| AIME | 36.7 | 11382 | 16.7 | 1172 | 53.3 | 5288 | 1.89 | 50.0 | 8821 | 1.31 | 6.7 | 4843 | 0.28 |
| Olympiad | 52.0 | 7762 | 38.2 | 944 | 50.7 | 3410 | 0.44 | 53.9 | 5544 | 0.40 | 29.6 | 3884 | 0.05 |
| AMC | 66.3 | 7414 | 51.8 | 1084 | 75.9 | 2580 | 1.09 | 72.3 | 4596 | 0.65 | 30.1 | 4304 | 0.31 |
| Minerva | 40.8 | 5034 | 38.2 | 1367 | 44.1 | 1341 | 0.98 | 40.4 | 2796 | 0.40 | 23.5 | 5277 | 0.55 |
| *Out-of-domain Tasks* | | | | | | | | | | | | | |
| MathQA | 83.3 | 3442 | 68.9 | 2585 | 85.4 | 980 | 0.79 | 78.8 | 1350 | 0.34 | 32.8 | 6321 | 0.14 |
| BBH | 68.3 | 2126 | 67.3 | 953 | 63.1 | 535 | 0.37 | 62.1 | 798 | 0.17 | 55.4 | 5431 | -0.35 |
| MMLU | 66.3 | 1528 | 76.0 | 269 | 68.6 | 567 | 0.73 | 69.3 | 891 | 0.55 | 72.3 | 5979 | 0.31 |
| LiveCodeBench | 50.6 | 8683 | 24.7 | 189 | 49.5 | 5228 | 0.29 | 49.7 | 7589 | 0.04 | 21.6 | 9726 | 0.20 |

Table 1: **Performance comparison of 7B models across diverse tasks.** For each task, we highlight the best-performing results across all evaluated models. The AE Scores are computed using hyperparameters $\varphi = 1$, $\eta = 3$, and $\theta = 5$. The proposed `ShorterBetter` model consistently achieves the highest AE Scores across both in-domain and out-of-domain benchmarks. For experimental details and additional results on coding tasks, refer to Appendix A and Appendix B.1.

as well as out-of-domain benchmarks.[5] We then provide an in-depth analysis of how our method changed the structure of the reasoning traces.

## 5.1 Experimental Results

**ShorterBetter significantly reduces inference length while maintaining performance** In Fig. 1, we compare the performance of the `ShorterBetter-1.5B` against `Distill-1.5B`, `Instruct-1.5B` and Training Efficient-1.5B. We report results for two variants of `ShorterBetter-1.5B`, with hyperparameters $\alpha = 1$ and $\alpha = 2$, respectively. In in-domain math tasks, averaging over all benchmarks, the two models achieve inference length reductions of 77.6% and 79.2%, with corresponding changes in accuracy of +2.5 % and +3.2 %, respectively. In out-of-domain tasks, averaging over all benchmarks, the two variants achieve average length reductions of 77.0 % and 75.5 % , with corresponding changes in accuracy of –0.8 % and –1.6 %, respectively.[6] These results also indicate that our method achieves similar performance in terms of length reduction and accuracy preservation with different values of correctness reward hyperparameter $\alpha$.

In Table 1, we provide a comprehensive comparison between `ShorterBetter-7B` ($\alpha = 2$) and four 7B baseline models.[7] Consistent with our 1.5B model evaluation, we assess all 7B models on both in-domain and out-of-domain tasks. For `Distill-7B` and `Instruct-7B`, we only report accuracy and output length as primary evaluation metrics since they didn't undergo length reduction training. For models trained with length-reduction techniques, we additionally report the AE Score, which quantifies performance improvements relative to their respective base models. (See a formal definition of AE Score in Appendix A.2)

Our 7B model achieves substantial reductions in output length, up to 70%, while maintaining competitive accuracy compared to the `Distill-7B` baseline. In in-domain math tasks, averaging over all benchmarks, `ShorterBetter-7B` attains a length reduction of 62.1% alongside an accuracy improvement of +7.1%. In out-of-domain tasks, averaging over all benchmarks, the model achieves a length reduction of 62.3% with an accuracy change of -0.48%. Moreover, `ShorterBetter-7B` attains the highest AE Scores in all evaluated tasks, outperforming Training Efficient and O1-Pruner.

Compared to prior length-reduction methods, `ShorterBetter` achieves stronger length-reduction capability. We attribute this performance to the design of SOL, which gives clear signals to guide the model to learn concise and robust reasoning traces during reinforcement learning. For more

---

[5]In-domain and out-of-domain tasks are defined relative to ShorterBetter's training dataset. See more discussions in Appendix A.3

[6]Due to space constraint, we only present LiveCodeBench here and place other coding benchmarks in Fig. 6 and Table 4 in the appendix

[7]For a fair comparison, we compute the AE Scores of O1-Pruner using its corresponding base model, Marco-o1-7B, whereas the base models for `ShorterBetter` and Training Efficient are both Distill-7B. Due to space limitations, the performance of Marco-o1-7B is placed in Appendix B.4

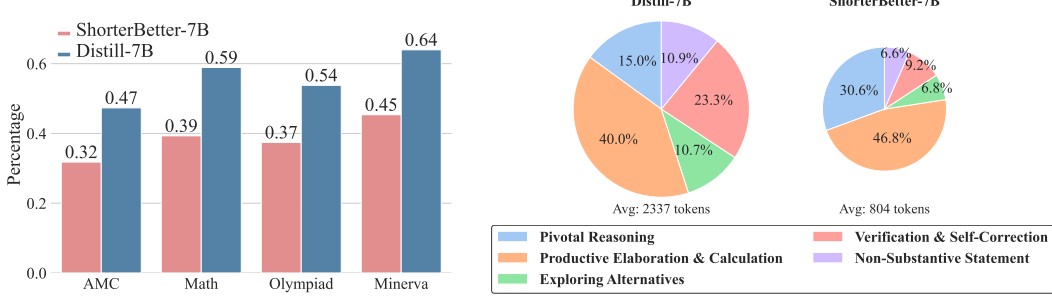

Figure 3: **Reasoning trace structure analysis. Left:** Percentage of remaining output after first appearance of the correct answer; **Right:** Token-level distribution of reasoning categories on the MATH benchmark.

discussions on the effectiveness of SOL, please refer to Appendix B.2. In the next section, we show that our method refines the structure of the reasoning traces relative to the base model.

## 5.2 Reasoning Structure Analysis

In this section, we present two novel frameworks for analyzing reasoning structure, which illustrate how `ShorterBetter` compresses its reasoning traces and complement the accuracy and token-length metrics introduced earlier. We employ Gemini 2.5 Flash Preview (Google, 2025) as an automated judge (details in Appendix E.1) to breakdown and compare the reasoning traces of `ShorterBetter-7B` and `Distill-7B`, shedding lights on how `ShorterBetter` optimizes for reasoning efficiency.

**Output Length after First Appearance of Correct Answer**  We use LLM-as-a-judge to identify the first occurrence of the correct answer in the reasoning trace for questions where both `ShorterBetter-7B` and `Distill-7B` provide correct answers. We conduct this analysis across four benchmarks: AMC, MATH, Olympiad, and Minerva. For each model, we calculate what percentage of its total output (in tokens) appears after the first time that the correct answer is stated. The results are shown in Fig. 3 (Left), where higher percentage represents more reasoning outputs after the model finds the correct answer. We observe that the percentages of tokens appearing after the first correct answer decreases by 15%, 20%, 17%, and 19% for AMC, MATH, Olympiad, and Minerva benchmarks, respectively. This suggests that `ShorterBetter-7B` reaches conclusions more efficiently, using fewer tokens on extra reasoning once it reaches the correct answer, and we will explore more detailed reasoning structure refinement in the next section.

**Breakdown of Reasoning Traces**  To better understand how `ShorterBetter` improves reasoning efficiency, we conduct a fine-grained analysis of the reasoning trace structure. Specifically, we develop an LLM-as-a-judge framework to assign functional categories to each sentence in the generated traces from `ShorterBetter-7B` and `Distill-7B`, and compute the token-level proportion of each reasoning category. We begin by carefully inspecting common patterns in Chain-of-Thought reasoning and curating a set of mutually exclusive categories, each assigned at the sentence level:

➢ **Pivotal Reasoning:** Steps that directly correspond to a specific part of the final solution.

➢ **Productive Elaboration & Calculation:** Necessary calculations, deductions, or planning that support a pivotal step but aren't restated in the final summary.

➢ **Exploring Alternatives:** Attempts to try different approaches or check other methods, even if they aren't used in the final solution.

➢ **Verification & Self-Correction:** Checks and corrections of earlier results to catch errors.

➢ **Non-Substantive Statement:** Redundant comments, filler, or trivial rephrasing that doesn't advance the solution.

We then use LLM-as-a-judge to label each sentence with the most appropriate category.[8] Finally, we compute the proportion of each category by counting tokens using the tested model's native tokenizer, enabling a consistent token-level comparison across models. In Fig. 3 (Right), we show the distribution of reasoning categories for both models on the MATH benchmark. Compared to `Distill-7B`, `ShorterBetter-7B` exhibits notable structural shifts. Specifically, the proportions of *Pivotal Reasoning* and *Productive Elaboration & Calculation* increase by 15.6% and 6.8%, respectively—indicating a denser concentration of meaningful, solution-oriented reasoning. Simultaneously, the share of *Non-Substantive Statements* decreases by 4.8%, suggesting that `ShorterBetter-7B` produces fewer filler or redundant tokens. We also observe reductions in *Verification & Self-Correction* (-14.1%) and *Exploring Alternatives* (-3.9%), implying that the model is more decisive and less prone to unnecessary detours. Altogether, these shifts support the conclusion that `ShorterBetter` promotes more confident and streamlined reasoning trajectories. This finding aligns with our analysis in the previous section and may help address the early stopping of promising paths issue mentioned in Section 1.

**Human Verification of LLM-as-a-judge**   In the reasoning-trace analysis above, we employed `Gemini 2.5 Flash` (Google, 2025) as a highly capable LLM judge, guided by a carefully designed, unambiguous evaluation prompt (see the prompts in Appendix E). To assess the reliability of its judgments, we conducted a comprehensive human verification study on the reasoning traces generated by `ShorterBetter-7B` for the AMC dataset. Specifically, we manually annotated 2,064 sentences across 49 problems (65,329 tokens in total), achieving 94.23% (1,945/2,064) agreement at the sentence level and 93.01% (60,765/65,329) at the token level with the LLM judge.

| Functional Category | LLM Judge | Human Judge |
| --- | --- | --- |
| Pivotal Reasoning | 22.37% | 23.42% |
| Productive Elaboration & Calculation | 53.32% | 50.79% |
| Exploring Alternatives | 10.50% | 10.79% |
| Verification & Self-Correction | 8.69% | 10.16% |
| Non-Substantive Statement | 5.13% | 4.84% |

Table 2: **Token-level distribution of reasoning categories judged by LLM versus human.** The result is aggregated on the reasoning traces generated by `Shorterbetter-7B` on the AMC dataset.

As shown in Table 2, inter-annotator agreement remains consistently high across all five categories, reinforcing the reliability of our LLM judge's categorizations. Thus, the reliability of our LLM judge's categorizations is strongly supported. Most disagreements occur between the categories "Productive Elaboration & Calculation" and "Pivotal Reasoning" – both regarded positively in our analysis. Consequently, our main conclusion – that `ShorterBetter` jointly increases the proportion of these meaningful reasoning categories – remains robust despite minor distributional discrepancies. We selected the AMC dataset for this validation study because it offers a balanced testbed: the problems require nontrivial mathematical reasoning while remaining tractable for exhaustive manual annotation. For transparency and to encourage reproducibility in LLM reasoning research, we have also released our annotation tool and human-labeled data in our open-source repository.

Overall, while our reasoning structure analysis in Section 5.2 provides valuable insights, we encourage future works to develop more rigorous quantitative metrics for evaluating reasoning structure, which could further validate and extend our findings.

### 5.3  Ablation Studies

**Naively adopting the shortest response as target length leads to training collapses**   We conducted three ablation experiments: (1) *No Correctness Reward, Shortest Length Target*: remove the correctness term in the reward function and always choose the length of the shortest response as the reward signal. (2) *Correctness Reward + Always Shortest Length Target*: we keep the correctness reward but always set the length target to the shortest response in a group. (3) *Correctness Reward + "Partial" Shortest Length Target*: when there is no correct response, select the length of the shortest

---

[8]For transparency and reproducibility, our prompt and examples of each category are provided in Appendix E.2

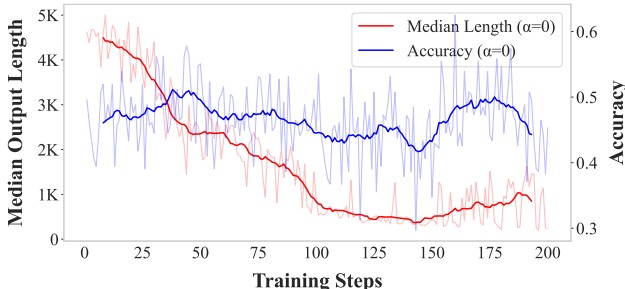

Figure 4: **Sample optimal length inherently combines correctness and conciseness.** Remove the correctness rewarding term in the reward function but keep the entire design of SOL, i.e., set $\alpha = 0$.

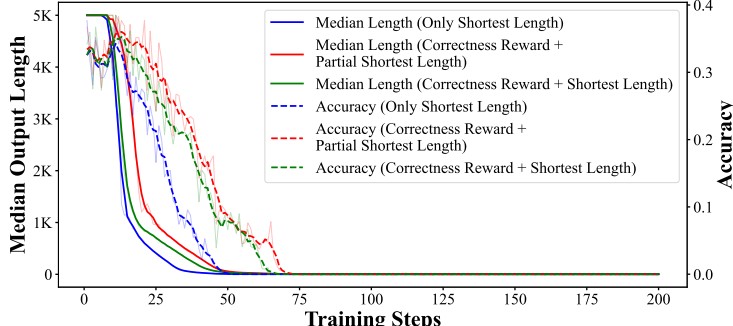

Figure 5: **Naively adopting the shortest response as target length leads to training collapses.** Ablation results show that replacing or partially replacing SOL with the shortest response length, with or without correctness reward, leads to training collapse within 100 steps.

response as the SOL, instead of using the average length as in Eq. (1).[9] In Fig. 5, model trainings under the three ablation experiment settings quickly collapse within 100 steps, suggesting that naively choosing the shortest response length as a reward can easily destabilize the training.

**Sample optimal length inherently combines correctness and conciseness**   We conducted another experiment that keeps the entire design of SOL while discarding the correctness reward term in the reward function, i.e., set $\alpha = 0$. As shown in Fig. 4, although there are drastic fluctuations in accuracy due to the absence of explicit correct rewards, the training still runs effectively. This result suggests that the design of SOL inherently combines correctness and conciseness. Nonetheless, incorporating a correctness reward term remains essential to prevent degradation in model accuracy.

## 6   Conclusion and Future Directions

This paper presents `ShorterBetter`, a reinforcement learning method that enables reasoning models to dynamically optimize their CoT lengths without manual supervision. By introducing Sample Optimal Length as a self-supervised reward signal, our method achieves significant reductions in output length while maintaining accuracy across in-domain and out-of-domain benchmarks. We also present innovative analytical framework to show `ShorterBetter`'s reasoning structure improvement.

While `ShorterBetter` is currently showing strong empirical results on tasks with *verifiable answers*, we note that our general formulation in Section 3.1 naturally extends to open-ended settings with non-binary correctness scores. Another future direction is to apply and evaluate our methods on larger-scale reasoning models (Team, 2024). Finally, our reasoning trace analysis shows that different categories of reasoning behavior are compressed to different extents, which may inform the design of more fine-grained and behavior-aware reasoning optimization strategies.

---

[9]Note that in experiment (1), we replace SOL with sample minimal length and discard the correctness term in the reward function. The reward function is rewritten as $r(y_j) = -\beta \left| \min_{y_i \in G(x_i)} \ell(y_i) - \ell(y_j) \right|$. While in experiment (2) and (3), we still keep the correctness term in the reward function and set $\alpha = 2$.

## Acknowledgment

We thank Data Science Institute at the University of Chicago for supporting our computational resources. We also thank Siyang Wu and Honglin Bao for their valuable feedbacks and suggestions on our earlier manuscript.

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

# ♣ Appendix: Table of Contents

## Broader Impact

This work contributes to the growing body of research on efficient reasoning in LLMs. By reducing redundant reasoning steps while maintaining accuracy, our method could help lower computational costs and improve the practicality of deploying LLMs in resource-constrained settings. Like many advancements in ML, this research may have indirect societal implications—for instance, more efficient models could reduce energy consumption or enable faster inference in applications such as education or automated decision-making. However, our focus is narrowly technical, and we do not make strong claims about specific real-world benefits or risks. As with any optimization technique, the broader impact will depend on how it is integrated into larger systems and deployed in practice.

## A  Technical and Experimental Details

### A.1  Detailed Specification of GRPO Algorithm

Here we provide the complete expressions and details of the GRPO (Guo et al., 2025) objective (loss function) mentioned in Section 3.3:

$$\mathcal{L}_{\mathrm{GRPO}}(\theta) = -\frac{1}{n} \sum_{j=1}^{n} \frac{1}{t_j} \sum_{k=1}^{t_j} \left[ W\big(\hat{A}_{j,k}, \, p_\theta^k, \, p_{\theta_{ref}}^k, \varepsilon\big) - \gamma \cdot \tilde{\mathbb{D}}_{\mathrm{KL}}\big[p_\theta^k \| p_{\theta_{ref}}^k\big] \right].$$

In particular, $W$ is a clipped policy improvement objective with full expression

$$W\big(\hat{A}_{j,k}, \, p_\theta^k, \, p_{\theta_{ref}}^k, \varepsilon\big) = \min\left( \frac{p_\theta^k}{p_{\theta_{ref}}^k} \hat{A}_{j,k}, \, \mathrm{clip}\left( \frac{p_\theta^k}{p_{\theta_{ref}}^k}; 1 - \varepsilon, 1 + \varepsilon \right) \hat{A}_{j,k} \right),$$

where the clipping function $\mathrm{clip}(x; a, b) := \min(b, \max(a, x))$ prevents the updated parameters from deviating too much from the reference policy. For KL divergence, Schulman (2020) proposes the following approximator

$$\tilde{\mathbb{D}}_{\mathrm{KL}}\big[p_\theta^k \| p_{\theta_{ref}}^k\big] := \frac{p_{\theta_{ref}}^k}{p_\theta^k} - \log \frac{p_{\theta_{ref}}^k}{p_\theta^k} - 1.$$

Finally, while the above formulation considers multiple rollouts for a single prompt $x_i$, in practice we compute the GRPO loss and update the policy parameters over a batch of $N$ prompts at each step.

### A.2  Detailed Formulation for Accuracy-Efficiency (AE) Score

The **Accuracy-Efficiency (AE) Score**, introduced by Luo et al. (2025a), provides a composite metric to evaluate whether a model improves inference efficiency—i.e., reduces output length—without sacrificing accuracy. It is defined as follows:

$$\text{AE Score} = \begin{cases} \varphi \cdot \Delta \text{Length} + \eta \cdot |\Delta \text{Acc}|, & \text{if } \Delta \text{Acc} \geq 0 \\ \varphi \cdot \Delta \text{Length} - \theta \cdot |\Delta \text{Acc}|, & \text{if } \Delta \text{Acc} < 0 \end{cases}$$

Here, $\Delta \text{Length}$ and $\Delta \text{Acc}$ denote the percentage changes in output length and accuracy, respectively, of the evaluated model relative to its base model. Positive AE Scores indicate that the model generates shorter outputs while maintaining or improving accuracy, while negative scores penalize accuracy degradation. In our experiments, we use the same hyperparameters as in Luo et al. (2025a):

- $\varphi = 1$ (weight on length reduction)
- $\eta = 3$ (bonus for accuracy gains)
- $\theta = 5$ (penalty for accuracy drops)

The asymmetric design (with $\theta > \eta$) penalizes accuracy drops more heavily than gains, reflecting the practical preference for avoiding performance degradation.

### A.3 Experimental Details

For evaluation, we categorize tasks as either in-domain or out-of-domain, based on the training data used for `ShorterBetter`. Our method is trained on the DeepScaleR-Preview dataset (Luo et al., 2025b), a composite benchmark that includes AIME, AMC, Olympiad-Bench, MATH, Minerva, etc. Accordingly, held-out subsets from these datasets are considered in-domain. In contrast, tasks such as MathQA, BBH, MMLU, LiveCodeBench, MBPP, and HumanEval—which are not included in the training corpus—are treated as out-of-domain.

Other length-reduction baselines are trained on different datasets. Specifically, Training Efficient (Arora and Zanette, 2025) is trained on MATH, CN-K12, AIME, AoPS, and Olympiad-Bench, while O1-Pruner (Luo et al., 2025a) is trained solely on MATH. There exists partial overlap in training data between these baselines and `ShorterBetter`. Nevertheless, the out-of-domain benchmarks defined in this work are held out from all models' training data and thus serve as a fair test of generalization.

For model comparisons in Fig. 1 and Table 1, we directly downloaded the open-source models of Training Efficient 1.5B/7B and O1-Pruner 7B from `HuggingFace` for evaluation. For O1-Pruner, since the paper does not provide a 1.5B version, we didn't include it in Fig. 1. For Training Efficient, we used the 1.5B and 7B versions both with hyperparameters $\alpha = 0.4$ (following the original paper). During all evaluations, the temperature was set to 0.9. All problems are answered once, and the accuracy is calculated as the fraction of correct answers among all answers.

### A.4 On the Practical Choice of Hyperparameters

**Choice of $\alpha$ and $\beta$** The hyperparameters $\alpha$ and $\beta$ are selected as a pair. Since reinforcement learning frameworks such as GRPO use relative rewards rather than absolute rewards, we fix $\beta = 0.001$ and focus on analyzing the choice of $\alpha$.

The hyperparameter $\alpha$ controls the trade-off between accuracy and efficiency. Its value relative to the length penalty $\beta$ (kept at 0.001) is the key factor. We tested new $\alpha$ values to illustrate its impact.

Table 3: Performance of ShorterBetter-1.5B with varying $\alpha$ values.

| Dataset | $\alpha = 0.1$ Acc | $\alpha = 0.1$ Len | $\alpha = 2.0$ Acc | $\alpha = 2.0$ Len | $\alpha = 5.0$ Acc | $\alpha = 5.0$ Len |
|---|---|---|---|---|---|---|
| AMC | 0.43 | 2503 | 0.57 | **1946** | **0.58** | 3077 |
| Minerva | 0.20 | 1785 | **0.28** | **1147** | 0.24 | 2249 |
| Olympiad | 0.30 | 2903 | **0.38** | **1814** | 0.35 | 3919 |
| AIME | 0.13 | 5395 | 0.20 | **2703** | **0.23** | 5274 |

Our results (shown in Table 3) provide an intuition for selecting $\alpha$:

- A high $\alpha$ (e.g., 5.0) places a strong emphasis on correctness. As seen in the training dynamics and the table above, this leads to stable accuracy gains but weaker length reduction, resulting in longer outputs compared to the more balanced setting.

- A low $\alpha$ (e.g., 0.1) makes the length penalty term negatively influential. This can degrade the model's reasoning capability, as the reward for correctness may not be sufficient to maintain performance. This instability can lead to worse length reduction because the model fails to learn effective, concise reasoning paths.

- A balanced $\alpha$ (e.g., 1.0 or 2.0) provides a strong enough correctness signal to preserve or improve accuracy while still exerting significant pressure to reduce output length. This achieves the best overall trade-off demonstrated in our paper.

In practice, the optimal choice depends on the specific goal (e.g., prioritizing efficiency vs. accuracy) and the specific model family. Based on our results, we recommend practitioners start with $\alpha$ in the range of [2, 5] and tune as needed.

## A.5  Computational Resources

We trained our `ShorterBetter-1.5B` with 4 A100 GPUs, using approximately 16 hours. We trained `ShorterBetter-7B` with 8 A100 GPUs, using less than 12 hours.

# B  Additional Experiment Results

## B.1  Coding Benchmarks

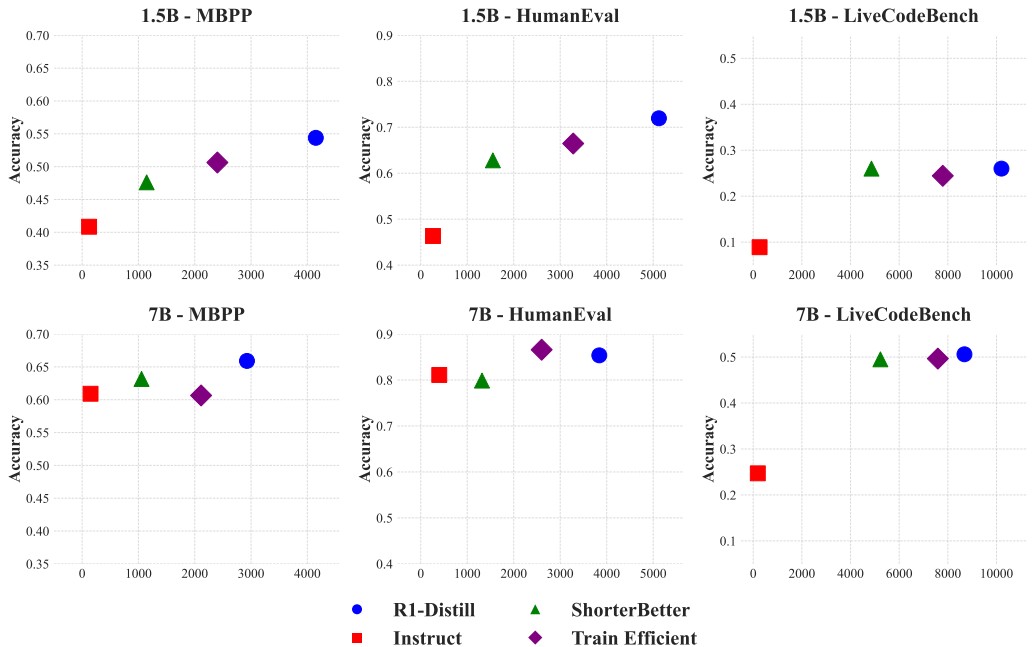

Figure 6: **Model performance on coding benchmarks.**

We report the result of the coding experiment in Table 4 and Fig. 6.

| | 1.5B Models | | | | 7B Models | | | | |
|---|---|---|---|---|---|---|---|---|---|
| Task | R1-Distill | Instruct | ShorterBetter | Training Efficient | R1-Distill | Instruct | ShorterBetter | Training Efficient | O1-pruner |
| MBPP | **0.544** (4151) | 0.409 (124) | 0.476 (1146) | 0.506 (2402) | **0.659** (2928) | 0.609 (150) | 0.632 (1055) | 0.607 (2113) | 0.494 (495) |
| HumanEval | **0.720** (5123) | 0.463 (263) | 0.628 (1551) | 0.665 (3280) | 0.854 (3841) | 0.811 (398) | 0.799 (1318) | **0.866** (2601) | 0.829 (254) |
| LiveCodeBench | **0.260** (10204) | 0.089 (262) | **0.260** (4861) | 0.244 (7792) | **0.506** (8683) | 0.247 (189) | 0.495 (5228) | 0.216 (9726) | 0.483 (1847) |

Table 4: Coding performance across model sizes and training methods. Each entry reports the accuracy rate and the average inference length in tokens (shown in parentheses). Both ShorterBetter models were fine-tuned with $\alpha = 2$. Baseline models are from Arora and Zanette (2025). Bold numbers indicate best accuracy for each model size category.

## B.2  Training dynamics of ShorterBetter

In Fig. 7 we show the output length (in tokens) trends and accuracy trends of `ShorterBetter-1.5B` and `ShorterBetter-7B` during GRPO training. For clearer illustration, we present the first 200 steps of the two models.

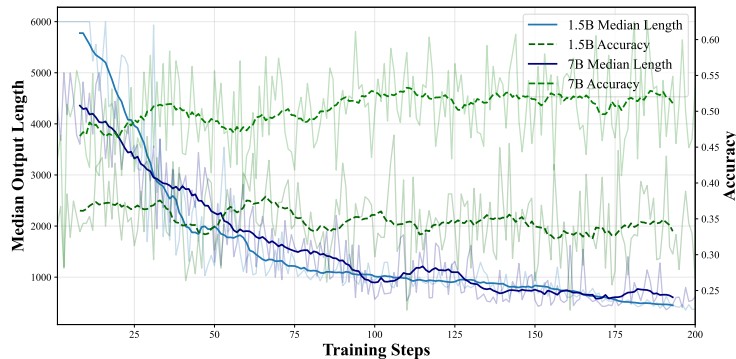

Figure 7: **Training dynamics of `ShorterBetter-1.5B` and `ShorterBetter-1.5B` models.**

In Fig. 8, we illustrate how the model learns to identify optimal inference lengths throughout the training. For each sample generation ($n = 8$) of a given problem, we compute the SOL and take the batch-wise average. We also track the mean deviation of individual output from their respective SOL, defined as $d_i = \frac{\ell(y_i) - \ell^{SOL}(G(x_i))}{\ell^{SOL}(G(x_i))}$. For instance, a deviation of 0.2 at a given training step indicates that the generated outputs deviate from the optimal length by 20% on average. The plots show that both the **SOL** and the mean deviation steadily decrease over training, particularly within the first 100 steps. This trend suggests that reinforcement learning is successfully guiding the model to produce outputs that cluster towards the optimal length.

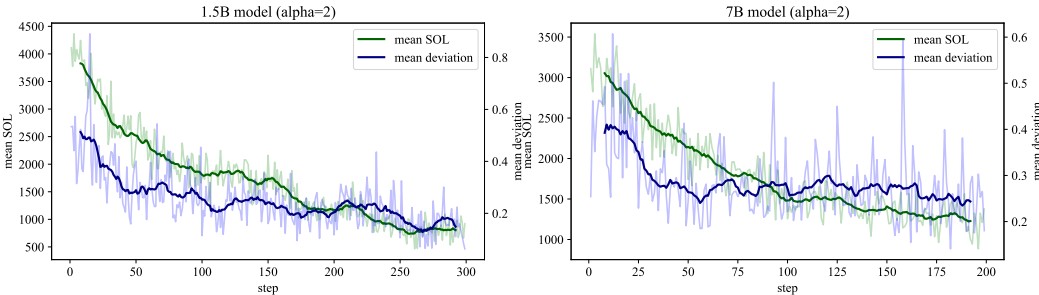

Figure 8: **SOL trends and mean deviations during training**

### B.3 Incorrect reasoning traces are significantly longer

In Fig. 9, we conducted an experiment using 150 problems (30 each from AIME, AMC, MATH, Olympiad-bench, and Minerva). For each problem, we generated 10 responses with the Distill-1.5B/7B models, then filtered out problems where all responses were correct or all responses were wrong. We then computed the difference between the average lengths of correct and incorrect responses to form a length gap distribution. The results show that, on average, for the same problem, incorrect responses are significantly longer than correct ones, by 5,541 tokens on DeepSeek-R1-Distill-Qwen-1.5B and 5,910 tokens on DeepSeek-R1-Distill-Qwen-7B. This suggests a clear correlation between reasoning errors and unnecessarily long CoT traces in Large Reasoning Models.

### B.4 Performance of Marco-o1

Here we report the performance of Marco-o1 on various benchmarks, which is used to calculate the AE Scores of O1-Pruner in Table 1.

### B.5 DeepSeek-R1-Distill-Llama-8B Results

To address model architecture bias, we evaluated `ShorterBetter` on a model with a different architectural backbone: DeepSeek-R1-Distill-Llama-8B. This allows us to test our method's generalization

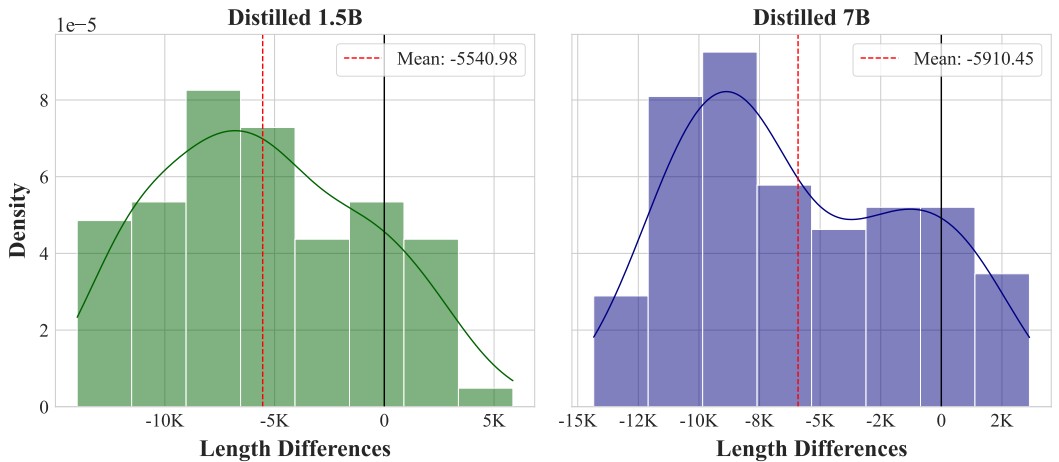

Figure 9: **Length gap distributions for the Distill-1.5B and 7B models.**

| Dataset | Accuracy | Length |
|---|---|---|
| AIME | 0.067 | 6711 |
| Olympiad | 0.317 | 6325 |
| AMC | 0.301 | 6210 |
| Minerva | 0.217 | 7583 |
| MathQA | 0.331 | 7730 |
| BBH | 0.623 | 6829 |
| MMLU | 0.701 | 7650 |
| LiveCodeBench | 0.225 | 12856 |
| HumanEval | 0.841 | 317 |
| MBPP | 0.454 | 877 |

Table 5: Marco-o1 performance across datasets

beyond the Qwen series. We trained this 8B Llama-based model for 100 steps using our framework. The results are shown in Table 6.

The results show that `ShorterBetter` successfully generalizes to a different model architecture. Here we choose hyperparameter $\alpha = 5$ since it achieves a better balance between accuracy and length reduction. However, the out-of-distribution performance has more deduction compared to Qwen series model, which suggests that fewer training steps or larger $\alpha$ might be needed.

## C  Extended Discussion on Related Work

Recent works such as `L1` (Aggarwal and Welleck, 2025) and `ConciseR` (Song and Zheng, 2025) share the goal of improving efficiency in reinforcement learning for reasoning, yet differ fundamentally from `ShorterBetter` in their problem formulations and design principles.

`L1` focuses on *external controllability*: a user provides an explicit reasoning-length budget, and the model is trained to adhere to this target. Consequently, the optimization objective and prompt structure are explicitly guided by human-specified constraints. In contrast, `ShorterBetter` introduces an *autonomous* mechanism that allows the model to *discover* its own optimal reasoning length based on the interplay between task difficulty and model capacity. This self-supervised setting removes the need for user-defined budgets, which may be unrealistic in scenarios where the efficient solution length is unknown *a priori*. Furthermore, while `L1` relies on prompt-level cues (e.g., "think for $n$ tokens"), `ShorterBetter` remains completely prompt-agnostic, enabling seamless application across diverse reasoning contexts.

|  | 8B-Distill | | ShorterBetter ($\alpha = 5$) | |
|  | Acc. | Len. | Acc. | Len. |
|---|---|---|---|---|
| *In-domain Tasks* | | | | |
| AIME | 0.33 | 9949 | **0.37** | **5634** |
| Olympiad | **0.45** | 7234 | 0.41 | **3944** |
| AMC | **0.67** | 6995 | 0.62 | **3384** |
| Minerva | **0.33** | 5555 | **0.33** | **2669** |
| *Out-of-domain Tasks* | | | | |
| MathQA | **0.77** | 3002 | 0.71 | **899** |
| BBH | **0.71** | 1860 | 0.60 | **699** |
| LiveCodeBench | **0.54** | 8101 | 0.37 | **1665** |

Table 6: **Performance comparison of 8B Llama-based models across diverse tasks.**

`ConciseR`, developed concurrently, employs a *two-stage gated optimization framework*: the first stage strengthens reasoning ability, and the second stage compresses reasoning length, with the conciseness reward activated only when all sampled responses are correct. This gating mechanism ensures correctness but limits sample efficiency, as full-batch correctness is rare in complex domains. By contrast, `ShorterBetter` adopts a *single unified learning process* that leverages the **Sample Optimal Length (SOL)**—the shortest correct reasoning trajectory among multiple generations—as a continuous, self-supervised reward signal. This design allows effective gradient propagation even when correctness is partial, substantially improving training efficiency and stability.

In summary, while all three methods operate within the emerging paradigm of reinforcement learning for efficient reasoning, `ShorterBetter` is distinguished by its integrated, self-supervised formulation and its introduction of **SOL** as a dynamic intrinsic reward. This contribution moves beyond explicit control (L1) and staged compression (`ConciseR`) toward an *autonomous discovery of efficiency*—capturing an intrinsic property of reasoning models rather than imposing it externally.

# D  Examples of Unwanted Reasoning Behaviors

## D.1  Verbose Repetition and Self-Verification in Intermediate Steps

---

**Prompt**

Denali and Nate work for a dog walking business and are paid for
each dog they walk.  Denali is responsible for $16$ dogs and Nate is
responsible for $12$ dogs.  Under the company's new policy, they will
be assigned or unassigned new dogs in groups of $x$ dogs.  The ratio
of Denali's pay to Nate's pay would be the same if Denali started
walking $4x$ more dogs and Nate stayed at $12$ dogs or if $x$ of Nate's
dogs were reassigned to Denali.  Find $x$ if $x \neq 0$.

---

**Response (DeepSeek-R1-Distill-Qwen-1.5B)**

...
Wait, perhaps the problem is not about the same pay ratio, but the
same total number of dogs?  But the problem says "the ratio of
Denali's pay to Nate's pay would be the same." So it's about the pay
ratio, not the number of dogs.
Wait, another thought:  Maybe the ratio is in terms of the number of
dogs walked, so if the ratio of their pay is same, and pay is per dog,
then perhaps the number of dogs walked should be proportional.
Wait, that is, if Denali walks 16 dogs and Nate walks 12, the ratio
of their pay is (16)/(12) if they are paid per dog.
...
Wait, so regardless of how I set up the ratio, I end up with x=0.
But since the problem states x≠0, perhaps my entire approach is
wrong.
Wait, let me think differently.
Perhaps the ratio of their pay is the same when they are assigned x
dogs, regardless of direction.
Wait, when they are assigned x dogs in groups of x, meaning x dogs
are reassigned.  So the ratio is same.
Perhaps it's the pay per dog ratio that's the same.
Wait, maybe Denali's pay per dog is same as Nate's.
But that would be same as each getting paid per dog.
Wait, but if they are reassigned, their pay per dog would be same.
Wait, perhaps we can think in terms of average.
Wait, I'm getting confused.  Maybe I should go back to the original
problem.
**Problem Statement:** ...
But we saw that this leads to x=0.
Alternatively, maybe the ratio is the other way around.
Wait, perhaps the ratio is the pay per dog ratio.  So if the pay per
dog is same, then the ratio is 1:1.
But in that case, both would have to walk same number of dogs.  But
the problem says that if x dogs are reassigned, it results in same
ratio.
But in that case, if the pay per dog is same, then it's just the same
number of dogs walked, which would imply that $16 - x = 16 + 4x \Rightarrow -5x =
0 \Rightarrow x = 0$, which is again invalid.
Wait, perhaps I need to approach this problem by assuming that the
pay ratio remains the same.
...

## D.2 Early Stopping of Promising Reasoning Paths

**Prompt**

The function $f$ satisfies the functional equation

$$f(x) + f(y) = f(x + y) - xy - 1$$

for all real numbers $x$ and $y$. If $f(1) = 1$, then find all integers $n$ such that $f(n) = n$. Enter all such integers, separated by commas.

**Response (DeepSeek-R1-Distill-Qwen-1.5B)**

```
...
Now, let's find f(2):
Using the recursive formula, set x = 1:
f(2) = f(1) + 1 + 2 = 1 + 1 + 2 = 4
Wait, is that right?  Let me check.
Wait, actually, according to the recursive formula f(x + 1) = f(x) +
x + 2, so for x = 1:  ...
```

We observe that the model starts to check for $x = 1$ immediately after it decides to check $f(2)$, interrupting its own verification step.

## D.3 Continued Exploration of Alternative Solutions After Reaching Correct Answer

**Prompt**

Find all values of $x$ that satisfy the equation $x = \sqrt{11 - 2x} + 4$.

**Response (DeepSeek-R1-Distill-Qwen-1.5B)**

```
...
So, the solutions are x = 1 and x = 5.  But wait, I need to check
if these solutions actually satisfy the original equation because
sometimes squaring both sides can introduce extraneous solutions.
First, let me check x = 1:
Left side:  1 = √11 - 2(1) + 4 Calculate inside the square root:  11 -
2 = 9 So, square root of 9 is 3.  Then 3 + 4 = 7 But left side is 1,
so 1 ≠ 7.  That means x = 1 is not a valid solution.
Now, check x = 5:
Left side:  5 = √11 - 2(5) + 4 Calculate inside the square root:  11
- 10 = 1 Square root of 1 is 1.  Then 1 + 4 = 5 So, 5 = 5, which is
true.  Therefore, x = 5 is a valid solution.
But wait, earlier I thought that x must be greater than or equal to 4.
Since x = 5 is greater than 4, that's fine.  But x = 1 is less than 4,
...
Wait, hold on.  Let me double-check my earlier factorization and
solutions.
I had x² - 6x + 5 = 0, which factored into (x - 1)(x - 5) = 0, giving x
= 1 and x = 5.
Plugging x = 5 into the original equation worked, but plugging x = 1
didn't.  So, x = 1 is extraneous.
But is there any other step I should verify?  Sometimes, especially
with square roots, squaring both sides can sometimes lead to other
issues, but in this case, since we didn't square the square root term
on both sides but isolated it once, it seems fine.
Let me consider another method to solve the equation, ...
```

```
...
Same quadratic as before, so same solutions.  So, same process, same
result.  So, x = 1 and x = 5, but only x = 5 is valid.
But just to make sure, let me consider if squaring the equation
introduced any extraneous solutions.  Sometimes when we square both
sides, it's easy to have extra roots.
...
```

## D.4 Collapse of Reasoning Coherence Following Very Long Outputs

```
Response (DeepSeek-R1-Distill-Qwen-1.5B; budget = 16000 tokens)

...
WaitWait endpoints.
about Wait.
, say needWait.
Maybe Wait Wait set's.
me Wait„ mark,:,'s No.
the„ me's number minimumButButWait.'s the,.
WaitWaitWait.
ao over be alone to..
, Wait.
.  the the the Wait Wait,.'s the me's Wait Wait the.'s.
me.  with of., range's., to (WaitWait k me nonWaitWait me's total
number integer's theBut a asagrams, No me„her.
Well Wait Wait a.  ...
```

# E Analyzing Reasoning Traces with LLM-as-a-judge

## E.1 LLM-as-a-judge Setup

In order to illustrate the reasoning structure improvement of SB, we used LLM-as-a-judge to help us analyze the reasoning trace because the lengthy reasoning trace is hard for human to read. We use Gemini 2.5 Flash Preview (Google, 2025) with top_p=0.95 and `temperature=0.7`. The prompt for reasoning trace categorization and identifying the first appearance of correct answer as follows:

---

**Prompt (Reasoning Categorization)**

You are a reasoning trace analyst.  Your role is to categorize each
line (separated by a newline, but do not treat equations as an
individual line) in a given reasoning trace according to the function
it serves within the reasoning process.  The goal is to understand
how different parts of the model's reasoning contribute (or not) to
the final answer.

You will be given a **full reasoning trace**, which ends with a final
answer after the `</think>`tag.  You must:

1.  **Only analyze the content before the `</think>`tag**.
2.  **Split the reasoning trace into individual lines**, multiple
sentences might be in a single line, a line should be ended by a
newline.
3.  **Assign exactly one label to each line** from the following
**mutually exclusive categories**.  Choose the **most fitting** label
based on the line's function in context.

### Categories:

1.  **Adopted Reasoning**
* **Definition**:  A reasoning step (e.g., setting up a key equation
that's shown in the summary, stating a critical definition that's
part of the summary, deriving a final value for a variable that's
highlighted in the summary, or an intermediate result explicitly
carried forward into the summary) whose content or conclusion is
**explicitly stated or directly corresponds to a distinct step,
equation, or piece of information in the summarized solution
presented after the `</think>`tag.** These are the landmark
achievements or key building blocks in the reasoning trace that are
showcased in the final, condensed explanation.
* *Example*:  If the final summary after `</think>`includes the
equation `$s^2+2s-11.25=0$`, then the line in the reasoning trace where
this equation is first derived or stated in this form is "Pivotal
Reasoning".  Similarly, the line stating "$s = 2.5$ km/h" is "Pivotal
Reasoning" if $s=2.5$ is a key reported value in the final summary.

2.  **Exploring Alternatives**
* **Definition**:  Lines that perform necessary calculations, logical
deductions, initial problem interpretation (e.g., "The problem asks
for...", "We are given..."), variable definitions (if not themselves
pivotal as per the above definition), strategic planning (e.g.,
"First, I will find s, then t."), or explanations of methodology that
**directly contribute to reaching a "Pivotal Reasoning" step or the
overall solution, but are not themselves explicitly restated as a
distinct element in the final summarized solution.**
* *Example*:   "Hmm, I need to convert t into hours because her

---

walking speed is in km/h." (Problem understanding leading to a pivotal setup). "Multiply both sides by $s(s+2)$: $18 = 1.6 \cdot s(s+2)$." (A calculation step leading to a pivotal equation). "Let me compute that left side: ... $= 18/(s(s\ +\ 2))$" (The process of algebraic simplification).

3. **Repetition & Rephrasing**
* **Definition**: Reasoning segments that explore solution paths, hypotheses, methods, or calculations that are **ultimately abandoned, identified by the LLM as incorrect or less viable, or otherwise do not form part of the sequence leading to the adopted final solution.** This includes trying a different formula that is then discarded, or starting a calculation that is then reversed due to a change in approach.
* *Example*: "Perhaps I can solve for $t$ first by isolating it from equation 1... no, subtracting the equations seems more direct to eliminate $t$." (The statement about solving for $t$ first by isolation would be "Alternative Path or Exploration" if that path is not taken).

4. **Verification & Explanation** * **Definition**: Lines that are redundant (e.g., repeating information or conclusions verbatim or near-verbatim shortly after they've been stated, without adding new purpose like emphasis for a next step), purely filler or conversational remarks (e.g., "Okay, so...", "Good.", "Let me think..."), or provide overly trivial explanations/rephrasing that do not advance the solution or significantly aid in understanding the problem-solving process (e.g., "This is a quadratic equation in s," immediately after writing $s^2 + 2s - 11.25 = 0$). This also includes restating parts of the problem verbatim without adding interpretation or planning elements, once the initial understanding is established.
* *Example*: "So, s is 2.5 km/h." (if stated immediately after "Thus, s = ... = 2.5 km/h" and not serving as a premise for an immediate subsequent calculation). "Let me write them down again:" (if the equations were just presented). "Yes, that seems correct," if it's a low-effort affirmation rather than the concluding part of a rigorous "Verification" step.

### Additional Instructions:

* **Stop processing as soon as you encounter the `</think>` tag**. Do not categorize anything beyond it. The content after `</think>`is used as the reference for "Pivotal Reasoning". * If a line **repeats or paraphrases** an earlier line without adding new value or serving a clear structural purpose (like summarizing inputs before a new phase of calculation), categorize the repeated instance as **"Non-Substantive Statement"**, even if the original served a different, valid purpose. * If a line could arguably fit into more than one category, choose the category that best describes its **primary function or most specific contribution** in that context. For instance, a calculation that corrects a previous error is "Verification & Self-Correction" rather than just "Productive Elaboration & Calculation." A statement of a key equation from the final summary is "Pivotal Reasoning" even if it also elaborates. * Do **not infer the logical correctness** of the reasoning or the final answer. The categorization is about the *structure and an LLM's perceived purpose* of each statement within its own reasoning

process. * Treat each line **independently for categorization**, but use the **surrounding context** (preceding and succeeding lines) to understand its function, especially for identifying repetitions, planning statements, or the flow of logic. * If the reasoning trace starts with the `</think>`tag, i.e.,
### Output Format:

Return your output as a **JSON array** of objects. Each object should have:

* "text": The full original line (string).
* "label": One of the five category names above (string).

```json [ "text": "Alternatively, we can try to factor the quadratic equation, but the roots might not be integers.", "label": "Exploring Alternatives" , "text": "Therefore, x $\bar{3}$ is the value we will use.", "label": "Pivotal Reasoning" , "text": "So, x is 3.", "label": "Non-Substantive Statement" ]asciigrave``

Return **only** the structured JSON list, no extra commentary or explanation.

---

**Prompt (Identify First Appearance of Correct Answer)**

You are a reasoning trace analyst. Your role is to identify the first sentence where the model gets the correct answer of the problem. The goal is to identify the redundant overthinking process after the model has actually solved the problem.

You will be given a full reasoning trace, which ends with a final answer after the `</think>`tag; you will also be given the problem and the correct answer. You must:

1. Only identify the sentence before the `</think>`tag.
2. Identify only the **first** sentence where the model gets the correct answer.
3. The sentence you return should be **exactly the same** as the one in the original reasoning trace.

Return only the sentence you identify, no extra commentary or explanation.

## E.2 Reasoning Category Examples

**Pivotal Reasoning Example**

First, I need to figure out the ratio of the enrollments between the two grades. The 7th grade has 520 students, and the 8th grade has 650. So, the total number of students is 520 + 650. Let me calculate that: 520 plus 650 is 1170. So, there are 1170 students in total between the two grades.

**Productive Elaboration & Calculation Example**

Therefore, the number of representatives for each grade should be proportional to their enrollment fractions. So, the 7th grade should have (4/9) of 18 representatives, and the 8th grade should have (5/9) of 18.

**Exploring Alternatives Example**

Hold on, maybe I need to think about the transversal lines and corresponding angles. Since ST is parallel to QR, and PR is a transversal, the angle at T, which is angle STR, should correspond to angle QRP, which is angle R, 105 degrees.

**Verification & Self-Correction Example**

Wait, if ST is parallel to QR, then the corresponding angles when PR is the transversal would be equal. So, angle S TR (which is angle STR) should be equal to angle QRP, which is 105 degrees.

**Non-Substantive Statement Example**

Wait, maybe I'm confusing the angles. Let me try to think of it differently.

