# OpenReview forum: "ShorterBetter: Guiding Reasoning Models to Find Optimal Inference Length for Efficient Reasoning"
_NeurIPS.cc/2025/Conference — NeurIPS 2025 poster_

### Official Review · Reviewer_5QGT · 2025-06-28

**Clarity:** 3
**Significance:** 2
**Originality:** 2
**Rating:** 4
**Confidence:** 2

**Summary:**

This study proposes a new method to reduce the length of the reasoning chain without sacrificing the model's performance. Specifically, in the proposed method, the shortest and correct reasoning chain is first determined via sampling, and then based on such statistics, the model reward is refined, and the model parameter is updated via Group Relative Policy Optimization. The proposed method shows more reduction in the output while maintaining the accuracy across broad benchmark sets.

**Questions:**

[Comparison with baselines]
The proposed method and baseline methods differ in several dimensions, such as training data,  hyperparameters (Appendix A.3), and formulation details, such as GRPO. Thus, it is not clear what was truly the key to achieving improvement over baselines. The use of out-of-domain seems a weak argument to skip such concerns (ll. 483–485).

Particularly, if the main claim of the paper is the use of the shortest correct response (as stated below and abstract), one should conduct clearer ablations about the definition of l^{SOL} (l.131).
>(Luo et al., 2025a), for example, compares generated lengths against those from a reference model. More in line with our proposal of sampling multiple responses, Kimi 1.5 (Team, 2025) and Training Efficient (Arora and Zanette, 2025) utilize the shortest and average lengths, respectively, among all sampled responses as reward baselines. In contrast, ShorterBetter innovatively anchors its reward function to the length of the shortest correct response

Specifically, one can examine the method variants without the correctness condition I(y_j=y_i*)=1 or just using the absolute length of the output in defining l^{SOL}.
Section 5.3 partially handles the concern around this point, but the analysis focuses more on orthogonal concerns about alpha and the case when there is no correct response.

[Experiments with other models]
I’m concerned about the possibility that the success of proposed methods is biased toward Deepseek-based models. These models yield somewhat distinctive outputs with, e.g., aha-moment, and I’m interested in whether the conclusions hold even with other models.

[Quality of shortened outputs]
Although coarse analysis on the reasoning structure has been done in Section 5.2, I'm more curious about the quality (human interpretability) of the shortened reasoning chains. If the shortened chain is no longer coherent, not helpful for human users (e.g., due to substantial omission in explanation), or amplifies inappropriate properties (e.g., hurting politeness), these will be limitations in a real human-LLM interaction. Such human-involved analysis will also be informative for the readers.

**Ethical Concerns:**

["NO or VERY MINOR ethics concerns only"]

**Final Justification:**

Through the rebuttal, my concerns, particularly on ablations, are (not perfectly but minimally) handled, which makes me lean toward acceptance (3->4).

**Limitations:**

yes

**Quality:**

2

**Strengths And Weaknesses:**

Strength
- The proposed approach achieved substantial output length reduction without compromising the accuracy
- Broad evaluation tasks, including out-of-domain ones, are adopted, and the advantage of the proposed method is consistently observed.
- Analysis of the reasoning structure is appreciated, which is informative about what kind of change is really made for the output


Weakness
- It is unclear which part of the proposed method truly contributed to the improvement over baseline methods, such as Training Inefficient (see Questions)
- The proposed method is only applied to DeepSeek-R1-Distill-Qwen-1.5B/7B models (see Questions)
- Further evaluation (or clarification) on the quality of the shortened reasoning chain (beyond correctness of the answer) will be encouraged (see Questions)

---

> ### Author Rebuttal · Authors · 2025-07-31
>
> We thank the reviewer for their constructive feedback. The questions raised about controlled comparisons, ablation clarity, model generalization, and quality of shortened outputs are crucial. We have conducted new experiments to address these points directly.
>
> ---
>
> ## **On the Comparison with Baselines**
>
> We agree with the reviewer that isolating the key factors for improvement is essential. To create a more controlled comparison and demonstrate the specific impact of our **Sample Optimal Length (SOL)** reward design, we conducted a new experiment.
>
> We implemented the reward function from a key baseline, `Training Efficient`, directly within our own training framework. This ensures all other variables are held constant:
> * **Base Model:** DeepSeek-Distill-Qwen-1.5B
> * **Training Data:** The same dataset used for `ShorterBetter`
> * **Training Algorithm:** GRPO
>
> The `Training Efficient` reward is formulated as $\mathbb{E}\left[\mathbb{1}_{y = y^*} \cdot (1 - \alpha f(\text{LEN}(y)))\right]$, where $f(\cdot)$ is a sigmoid function over the normalized length. We used $\alpha=1$ for the most aggressive length reduction setting.
>
> **Controlled Comparison on Math Benchmarks (1.5B Models)**
>
> | Dataset | `ShorterBetter` Acc. | `ShorterBetter` Len. | `Training Efficient` Reward Acc. | `Training Efficient` Reward Len. |
> | :--- | :---: | :---: | :---: | :---: |
> | AMC | **0.57** | **1946** | 0.49 | 2032 |
> | Minerva | **0.28** | **1147** | 0.23 | 1472 |
> | Olympiad | **0.38** | **1814** | 0.35 | 2597 |
> | AIME | **0.20** | **2703** | 0.20 | 5219 |
>
> The results show that under identical training conditions, our SOL-based reward function outperforms the baseline's formulation in both accuracy and length reduction across nearly all benchmarks. This provides strong evidence that our reward design is a key driver of the observed improvements.
>
> Of course, we acknowledge that a full "apple-to-apple" comparison with `Training Efficient` is unfortunately infeasible within the rebuttal window, because our settings differ in data, RL algorithm, reward design, and compute requirements. Here, we summarize the key distinctions:
>
> | Aspect               | `Training Efficient`                       | `ShorterBetter`                                              |
> | -------------------- | ---------------------------------------- | ---------------------------------------------------------- |
> | RL method            | PPO                                      | GRPO (group‑based)                                         |
> | Reward function      | Normalized length + sigmoid              | SOL                                                        |
> | Data                 | MATH, cnk12, AIME, AoPS and the Olympiad | DeepScaleR                                                 |
> | Compute requirements | 7B & 1.5B models: 8 × GB200 for 20 h     | 7B model: 8 × A100 for 10 h; 1.5B model: 4 × A100 for 16 h |
>
> In the paper, we compared our trained model with their released checkpoints. `Training Efficient` shows weaker length reduction, which we believe is due to its conservative reward design: the length signal is normalized and passed through a sigmoid, so long but correct answers are still rewarded, just slightly less. They did not provide the model with a clear signal during RL.
>
> However, `ShorterBetter` defines an explicit pivot (SOL) and lets it guide the model to quickly identify the desired output length. For example:
>
> - Suppose a prompt yields 8 responses, and 2 are correct.
> - One correct answer is **length 100**, another is **length 1000**.
> - `Training Efficient`: Both are rewarded positively, with the longer one slightly less.
> - `ShorterBetter`: The 1000-token answer is **penalized**, because the 100-token answer proves the task can be solved concisely.
>
> This design allows the model to quickly converge toward short, correct answers and achieve higher efficiency in length reduction without sacrificing accuracy.
>
> ---
>
> ## **On Clearer Ablation of the Reward Function**
>
> We agree that a clear ablation on the definition of $l^{SOL}$ is essential to prove our central claim. As the reviewer suggested, we tested variants that remove or alter the correctness condition. Our ablation studies confirm that naively targeting the shortest length is not viable and that the specific design of SOL is critical.
>
> 1.  **No Correctness Reward, Shortest Length Target:** As shown in Section 5.3, if the reward only minimizes length by targeting the shortest response in the group, the model quickly learns to produce empty outputs, causing accuracy to collapse to zero.
> 2. **"Partial" Shortest Length Target:** Using the shortest length as a target *only* when no correct answers are found also fails. This creates a feedback loop where the model sacrifices correctness for length, leading to training collapse.
> 3.  **Correctness Reward + Always Shortest Length Target:** In a new ablation, we kept the correctness reward ($\alpha=2$) but always set the length target to the shortest response in a group. This also crashed the training, with logs showing accuracy and length rapidly declining to zero. Even with a reward for correctness, a single short (but incorrect) response heavily penalizes all longer (and potentially correct) responses, creating a fatal bias toward underthinking.
>
> These experiments underscore that the length target design is paramount. Our proposed **Sample Optimal Length (SOL)**, which is robustly anchored to the shortest *correct* response and defaults to the *average* length in failure cases, provides the necessary stability to avoid these collapse scenarios.
>
> ---
>
> ## **On Experiments with Other Models**
>
> To address the valid concern about model architecture bias, we evaluated ShorterBetter on a model with a different architectural backbone: **`deepseek-ai/DeepSeek-R1-Distill-Llama-8B`**. This allows us to test our method's generalization beyond the Qwen series. We trained this 8B Llama-based model for 100 steps using our framework.
>
> | Dataset | Baseline (Llama-8B) Acc. | Baseline (Llama-8B) Len. | SB alpha=5  (Llama-8B) Acc. | SB alpha=5  (Llama-8B) Len. | SB alpha=2  (Llama-8B) Acc. | SB alpha=2  (Llama-8B) Len. |
> | :--- | :---: | :---: | :---: | :---: |  :---: | :---: |
> | AMC | **0.67** | 6995 | 0.62 | 3384 | 0.57 | **2808** |
> | Minerva | **0.33** | 5555 | **0.33** | 2669 | 0.30 | **1343** |
> | AIME | 0.33 | 9949 | **0.37** | **5634** | 0.27 | 6060 |
>
> The results show that `ShorterBetter` successfully generalizes to a different model architecture. Here we choose hyperparameter alpha=5 since it achieves a better balance between accuracy and length reduction.
>
> Regarding the "aha-moment" outputs, we view this as a characteristic of certain reasoning trace cases generated during inference, rather than a fundamental property of the DeepSeek distilled model series that would uniquely bias our method's effectiveness. The results on the Llama model support the conclusion that our method is broadly applicable.
>
> ---
>
> ## **On Quality of Shortened Outputs**
>
> We thank the reviewer for highlighting the importance of human-involved analysis regarding the quality and interpretability of shortened reasoning traces. To directly address this, we conducted a detailed manual analysis of `ShorterBetter-7B`’s reasoning traces for all 46 problems from the AMC dataset. Our analysis yields two primary insights:
>
> 1. **Coarse-level human agreement:** As detailed in our response to Reviewer 3dRz, inter-annotator agreement between humans and our LLM judge is high (sentence-level category accuracy: 94.23%). This supports the reliability of our reasoning structure analysis framework.
>
> 2. **Preservation of coherence and readability:** Through careful inspection, we confirmed that the coherence of reasoning traces remains largely intact. Specifically, consider the first problem in the AMC dataset:
>
> > *$\frac{m}{n}$ is the irreducible fraction value of $3+\frac{1}{3+\frac{1}{3+\frac{1}{3}}}$. What is the value of $m+n$?*
>
> Both baseline and `ShorterBetter-7B` models answered correctly, but ShorterBetter reduced reasoning length from 1320 to 654 tokens. Human inspection revealed some key insights:
>
> - **Removal of non-substantive statements:** Unnecessary statements (e.g., "I think that's solid" or "each step seems to check out") were effectively eliminated.
> - **Reduction in redundant verification:** The baseline explicitly and verbatim repeated calculations. ShorterBetter instead employed more concise verification methods, utilizing assumptions inherent in the problem.
> - **More concise pivotal reasoning:** ShorterBetter replaced overly verbalized calculations (e.g., "So now, adding $\frac{9}{3}$ + $\frac{1}{3}$ gives me $\frac{10}{3}$") with succinct mathematical expressions, preserving clarity and coherence.
>
> Overall, we observe that `ShorterBetter`’s RL fine-tuning shrinks reasoning trace length without compromising readability and interpretability. In fact, its reductions are very reasonable and human understandable.
>
> We acknowledge that our current evaluation primarily addresses rigorously structured domains (e.g., mathematics & programming), where criteria such as "politeness" or "helpfulness" are less pertinent. Extending our framework to open-ended settings may introduce new challenges—such as potential reward hacking—where reducing trace length might negatively impact such human-interaction qualities. We plan to investigate these issues in future work.

---

> > ### Comment · Reviewer_5QGT · 2025-08-04
> >
> > I appreciate author responses to handle my concerns. Based on the results of the reward ablation, results with another model, and the implications from manual analysis, my concerns have been generally handled, and I'd like to raise my score.
> >
> > Addressing the following points will further enhance the quality of this work, and I'd like to see these results in the updated version:
> > - The reward ablation is only done with 1.5B model during author response, which immediately encourages the same ablation with the 7B models
> > - The inclusion of Deepseek-R1-Distill-Llama-8b is minimally okay, but this still shares the same distillation process with DeepSeek-R1-Distill-Qwen models; both may have the same biases inherited from Deepseek. Ideally speaking, it's better to include completely different models, such as vanilla LLaMa models.
> > -  As raised by other reviewers (particularly, Reviewer iAtn), the difference with other methods should be more clearly explained, which will be good to introduce this field to a broad audience.

---

> > > ### Author Response · Authors · 2025-08-05
> > >
> > > We thank the reviewer for their positive comments and for raising their score. We will incorporate the new experiments and discussions from our rebuttal into the updated version. This includes the controlled comparison with a key baseline under identical training conditions, a more detailed ablation of our reward function's design (with 7B models), and the results from applying our method to a different model architecture. We will also provide a more detailed discussion on the comparison with other methods to better contextualize our contributions.

---

### Official Review · Reviewer_3dRz · 2025-07-03

**Clarity:** 3
**Significance:** 2
**Originality:** 2
**Rating:** 4
**Confidence:** 3

**Summary:**

This paper introduces "ShorterBetter," a reinforcement learning method designed to mitigate the "overthinking" phenomenon in Large Reasoning Models. The core contribution is a novel, self-supervised reward mechanism based on the concept of "Sample Optimal Length" (SOL). For a given problem, the model generates a small batch of responses, and the SOL is defined as the token length of the shortest correct response within that batch. If no response is correct, the average length of the incorrect responses is used as a neutral baseline. This SOL-based reward is then used to fine-tune the LRM using the GRPO algorithm, guiding the model to produce more concise yet accurate reasoning traces without requiring manual length specifications. The authors demonstrate empirically that applying ShorterBetter to 1.5B and 7B parameter models results in a 50-80% reduction in output length on both in-domain and out-of-domain benchmarks, while largely maintaining task accuracy.

**Questions:**

refer to weakness

**Ethical Concerns:**

["NO or VERY MINOR ethics concerns only"]

**Limitations:**

refer to weakness

**Quality:**

2

**Strengths And Weaknesses:**

Strengths
This paper presents a compelling and timely solution to a significant problem in modern LRMs. Its primary strengths lie in its originality, significance, and strong empirical results. The proposed method is novel in its use of SOL as a dynamic, data-driven reward signal, offering an elegant alternative to methods requiring explicit length budgets. The problem of computational inefficiency from "overthinking" is highly significant for the practical application of LRMs, and this work provides a promising path toward more efficient and deployable models. The quality of the empirical work is high; the reported length reductions are substantial and consistently demonstrated across multiple models and diverse benchmarks, supported by ablation studies that validate key design choices. The paper is also written with excellent clarity, making the methodology and results easy to follow.

Weaknesses

The robustness of the core method is questionable due to its reliance on a small sample size. The entire reward signal hinges on the SOL, which is derived from a small group of n=8 rollouts per prompt during training. This small sample size can lead to a high-variance, unstable reward signal. The calculated SOL in any given step might be an artifact of a stochastic sampling outcome rather than a true reflection of the model's optimal reasoning capability for a given problem. If a truly concise and correct reasoning path is not among the 8 samples, the model will be guided toward a suboptimal target. The paper lacks a crucial sensitivity analysis exploring how the choice of n impacts training stability and final model performance. This omission undermines the quality and robustness of the proposed method, as its effectiveness may be highly dependent on this unexamined hyperparameter.

A key claim of the paper is that ShorterBetter refines the structure of reasoning traces, not just their length. However, the evidence for this claim is based on an analysis performed by another LLM (Gemini 2.5 Flash Preview) acting as an automated judge. The authors use this LLM-as-a-judge to categorize sentences into functional roles like "Pivotal Reasoning" or "Exploring Alternatives" and then report shifts in their token-level proportions. The critical flaw in this evaluation is the lack of any validation for the judge's accuracy or objectivity. The paper does not report inter-annotator agreement between the LLM judge and human experts, nor does it provide any metrics to quantify the reliability of these automated categorizations. Without such validation, the conclusions about structural improvements are built on an unverified and potentially biased foundation, weakening the quality and significance of this portion of the paper's contribution.

---

> ### Author Rebuttal · Authors · 2025-07-30
>
> ## **On the choice of rollout size $n$**
>
> We appreciate the reviewer for highlighting the ambiguity surrounding our choice of rollout size n. Here, we clarify how the rollout size *n* affects our GRPO algorithm, particularly when *n = 4, 8,* and *12* on the 1.5 B model. (Results for *n = 4* and *n = 12* are new and were not included in the original paper.)
>
>
> *Computational constraints*
>
> * A larger *n* increases GPU memory and wall-time requirements, especially in early training when outputs are still long.
> * With large *n* the agent may exhaust memory and trigger OOM errors; therefore we restrict experiments to modest *n* values.
> * In practice we were able to run *n = 4, 8,* and *12* reliably; higher values were infeasible on our hardware.
>
> *Training dynamics*
>
> #### Due to limited time and computational budget, we were unable to run a full evaluation suite for the rollout size variants (n = 4, 8, 12). Nonetheless, the training-dynamics already capture how each choice of n affects convergence speed, length reduction, and accuracy stability, thereby providing the key evidence needed to understand their impact. All training-dynamics metrics are tested on questions unseen by the model, thereby faithfully reflecting the generalisation performance and output efficiency.
>
> Notes: Due to NeurIPS 2025 formatting constraints, we are unable to embed the training-dynamics plots in this rebuttal.
> | Rollout *n* | Accuracy trend                                                   | Length-reduction behaviour                                                                                        | Interpretation                                                                                                |
> | ------------ | ---------------------------------------------------------------- | ----------------------------------------------------------------------------------------------------------------- | ------------------------------------------------------------------------------------------------------------- |
> | **4**        | Stable, no noticeable oscillations                               | Converges slowly; at step 100 the median length remains **> 1000 tokens**, and later steps rarely shorten further | Sampling space too narrow; once answers reach a moderate length the policy lacks diversity to push shorter    |
> | **8**        | Moderate, well-controlled fluctuations (see Fig. 6 in the paper) | Good balance: faster length drop than *n = 4* while maintaining accuracy                                          | Trade-off point where learning is both efficient and stable                                                   |
> | **12**       | Large oscillations with a downward accuracy trend                | Very fast: by **step 40** median length falls **< 1000 tokens**; by **step 200** it reaches **≈ 300 tokens**      | Wide sampling space produces aggressive SOL targets, accelerating length shrinkage but destabilising accuracy |
>
> * **Key observation:** Increasing *n* enlarges the candidate pool, so the chosen SOL target becomes more aggressive. This markedly accelerates length reduction but also amplifies accuracy variance.
> * **Practical recommendation:** *n = 8* offers the best balance between rapid convergence and accuracy stability and, critically, fits within the resource budget for larger models such as 7 B parameters.
>
> ---
>
> ## **On the validation of automated judge:**
>
> We thank the reviewer for highlighting the importance of validating our automated LLM judge. To address this, we conducted a comprehensive human-annotation study on the AMC dataset (46 problems, 2064 sentences, 65,329 tokens) using `ShorterBetter-7B`’s reasoning traces. Key inter-annotator validation results are as follows:
>
> * **Sentence-level accuracy:** 94.23% (1945/2064)
> * **Token-level accuracy:** 93.01% (60,765/65,329)
>
> The token-level distribution comparison between the LLM and human judges is summarized below:
>
> | Functional Category                  | LLM Judge | Human Judge |
> | ------------------------------------ | --------- | ----------- |
> | Pivotal Reasoning                    | 22.37%    | 23.42%      |
> | Productive Elaboration & Calculation | 53.32%    | 50.79%      |
> | Exploring Alternatives               | 10.50%    | 10.79%      |
> | Verification & Self-Correction       | 8.69%     | 10.16%      |
> | Non-Substantive Statement            | 5.13%     | 4.84%       |
>
> Qualitatively, we observe that:
>
> * The inter-annotator agreement is consistently very high across all five categories. Thus, the reliability of our LLM judge’s categorizations is strongly supported.
> * Most disagreements occur between the categories "Productive Elaboration & Calculation" and "Pivotal Reasoning"—both regarded positively in our analysis. Consequently, our main conclusion—that ShorterBetter jointly increases the proportion of these meaningful reasoning categories—remains robust despite minor distributional discrepancies.
>
> We selected the AMC dataset as it strikes an ideal balance: it features non-trivial mathematical reasoning while remaining small enough to annotate exhaustively within our rebuttal timeline. For transparency, we plan to release our human-annotation platform and extend validation to additional benchmarks in the near future.

---

> > ### Comment · Reviewer_3dRz · 2025-08-07
> >
> > Thanks for the detailed rebuttal. I'll keep my score.

---

### Official Review · Reviewer_2UEk · 2025-07-03

**Clarity:** 4
**Significance:** 3
**Originality:** 3
**Rating:** 4
**Confidence:** 4

**Summary:**

Reasoning models tend to generate redundant and inefficient chain-of-thought (CoT) traces for complex tasks, a phenomenon commonly known as overthinking. The paper proposes a simple yet effective reinforcement learning method to enable models to learn their own optimal CoT lengths without manual supervision. The length of the shortest correct response among multiple generations is used as a dynamic reward signal to guide model training. The proposed method has been shown to be able to reduce unnecessary repetitions, excessive self-verification, and over-exploration of alternatives, achieving significant reduction in output length while maintaining reasoning accuracy.

**Questions:**

-	Can ShorterBetter address the issues when the sampled response is still long or encourages taking reasoning shortcuts that adversely affect out-of-distribution generalization?
-	Are there any tradeoffs in determining the hyperparameters of the proposed method?

**Ethical Concerns:**

["NO or VERY MINOR ethics concerns only"]

**Final Justification:**

I thank the authors for the additional experiments on the hyperparameters, which are helpful, and I encourage them to include the results in the paper.

While I still have some concerns about potential reasoning shortcuts introduced by ShorterBetter, I will retain my original rating to show my support for this work.

**Limitations:**

yes

**Quality:**

3

**Strengths And Weaknesses:**

**Strengths**

-	The proposed method is effective in reducing the length of reasoning traces while being able to maintain model accuracy. The obtained models perform well on both in-domain and out-of-domain benchmarks with reduced reasoning lengths.
-	The proposed reasoning trace analysis frameworks, including analyzing the length of extra reasoning after reaching correct answers and fine-grained analysis of the reasoning trace structure, are useful for this line of research.
-	The proposed sample optimal length serves as a dynamic and effective reward for learning efficient reasoning models.
-	The paper is well-written and easy to follow.

**Weaknesses**

-	The proposed method, ShorterBetter, is motivated from the optimal reasoning length hypothesis. To approximate the optimal length, the method adopts a sample-based approach to select the shortest length of correct responses from $n$ generated rollouts. It is possible that the selected response is still long. It is also possible that the selected response is too short, so the model takes some reasoning shortcuts, resulting in reduced performance on out-of-distribution benchmarks.  It is unclear how ShorterBetter can address these issues.
-	There are three important hyperparameters in  ShorterBetter, i.e., $\alpha$, $\beta$, and $n$. There lacks a discussion on how these parameters are chosen and what are the tradeoffs for setting these parameters.

---

> ### Author Rebuttal · Authors · 2025-07-31
>
> We thank the reviewer for highlighting the ambiguity in our interpretation of SOL and in the choice of hyperparameters. In response to your questions, our brief replies are as follows:
> 1. Our method naturally handles both long responses and avoids harmful reasoning shortcuts by adaptively balancing correctness and length.
> 2. The main trade-off lies in tuning $\alpha$ and rollout size $n$, which balance accuracy and output length.
>
> Detailed analysis and supporting experiments are provided below.
>
> ---
>
> ## **On the concerns on SOL design**
>
> **1. SOL might be too long**
> - A long SOL typically arises only on genuinely difficult problems.
> - In such cases the shortest correct answer must necessarily be long, so an aggressive length penalty would be inappropriate.
> - Our reward function therefore behaves *conservatively* when SOL is large; it can even penalise outputs that are *too short*.
> - This behaviour is desirable: the model learns to respect problem difficulty and produce longer reasoning chains when required.
> - Empirically, Fig. 1 and Table 1 confirm this dynamic balance: after training, outputs on harder benchmarks such as **AIME** and **LiveCodeBench** remain noticeably longer than on easier sets, even though overall lengths are still reduced.
>
>
> **2. SOL might be too short, causing reasoning shortcuts**
>
> We manually inspected ~50 training examples to test whether the model was merely “guessing” the correct answer with an incorrect process (you can also refer to the last part of our reply to reviewer 5QGT). Such cases were **rare**:
>
> * All training questions are *not* multiple-choice, so random guessing is extremely unlikely.
> * For datasets like **AIME** (answer ∈ [1, 100]) a small chance of lucky guesses exists, but its impact on training is negligible.
> * Moreover, the **α** coefficient in the correctness-reward term dampens accuracy oscillations: if an overly short SOL starts to harm reasoning quality, the α-weighted reward for correctness pulls training back on track.
> * In later training stages—once lengths have already been markedly reduced—the relative weight of **α** in the reward becomes larger, effectively stabilising any residual accuracy fluctuations.
> * However, we did observe that very large rollout sizes (*n*) occasionally produce *over-aggressive* SOLs, leading to accuracy oscillations. Hence, we should choose a moderate *n* (see detailed analysis below) to avoid this issue.
>
> ---
>
>
> ## **On the Practical Choice of Hyperparameters**
>
> **On the choice of $\alpha$ and $\beta$**
>
> $\alpha$ and $\beta$ are selected as a pair. Since reinforcement learning frameworks such as GRPO use **relative rewards** rather than **absolute rewards**, we fix $\beta = 0.001$ and focus on analyzing the choice of $\alpha$.
>
> The hyperparameter $\alpha$ controls the trade-off between accuracy and efficiency. Its value relative to the length penalty $\beta$ (kept at 0.001) is the key factor. We tested new $\alpha$ values to illustrate its impact.
>
> **Performance of `ShorterBetter-1.5B` with Varying $\alpha$**
>
> | Dataset  | $\alpha$=0.1 Acc | $\alpha$=0.1 Len | $\alpha$=2.0 Acc | $\alpha$=2.0 Len | $\alpha$=5.0 Acc | $\alpha$=5.0 Len |
> | :--- | :---: | :---: | :---: | :---: | :---: | :---: |
> | AMC      | 0.43         | 2503      | 0.57         | **1946** | **0.58** | 3077      |
> | Minerva  | 0.20         | 1785      | **0.28** | **1147** | 0.24         | 2249      |
> | Olympiad | 0.30         | 2903      | **0.38** | **1814** | 0.35         | 3919      |
> | AIME     | 0.13         | 5395      | 0.20         | **2703** | **0.23** | 5274      |
>
>
> We present new results for $\alpha=0.1$ and $\alpha=5.0$ to complement the results for $\alpha \in \{0, 1, 2\}$ in the main paper.
>
> For alpha = 0.1, the training log shows that accuracy follows a declining trend over 200 training steps. During this time, the median output length decreases rapidly.
> For alpha = 5.0, the training log indicates a clear upward trend in accuracy throughout the training process. The median output length also trends downward, decreasing from 5000 to below 2000 tokens.
>
> Our findings provide an intuition for selecting $\alpha$:
>
> * **A high $\alpha$ (e.g., 5.0)** places a strong emphasis on correctness. As seen in our training dynamics log and the table above, this leads to stable accuracy gains but a weaker length-reduction, resulting in longer outputs compared to the more balanced setting. The accuracy trend is clearly positive during training.
> * **A low $\alpha$ (e.g., 0.1)** makes the length penalty term negatively influential. This creates a risk of degrading the model's reasoning capability, as the reward for correctness may not be sufficient to maintain performance. The training dynamics log shows a downward trend in accuracy. This instability can lead to worse length reduction, as seen in the table, because the model fails to learn effective, concise reasoning paths.
> * **A balanced $\alpha$ (e.g., 1.0 or 2.0)** provides a strong enough correctness signal to preserve or improve accuracy while still exerting significant pressure to reduce output length. This achieves the best overall trade-off demonstrated in our paper.
>
> In practice, the optimal choice depends on the specific goal (e.g., prioritizing efficiency vs. accuracy) and specific model family. Based on our results, we recommend practitioners start with $\alpha$ in the range of [2, 5] and tune as needed.
>
> **On the choice of rollout size $n$**
>
> Here, we clarify how the rollout size *n* affects our GRPO algorithm, particularly when *n = 4, 8,* and *12* on the 1.5 B model. (Results for *n = 4* and *n = 12* are new and were not included in the original paper.)
>
> *Computational constraints*
>
> * A larger *n* increases GPU memory and wall-time requirements, especially in early training when outputs are still long.
> * With large *n* the agent may exhaust memory and trigger OOM errors; therefore we restrict experiments to modest *n* values.
> * In practice we were able to run *n = 4, 8,* and *12* reliably; higher values were infeasible on our hardware.
>
> *Training dynamics*
>
> #### All training-dynamics metrics are tested on questions unseen by the model, thereby faithfully reflecting the generalisation performance and output efficiency.
>
> | Rollout *n* | Accuracy trend                                                   | Length-reduction behaviour                                                                                        | Interpretation                                                                                                |
> | ------------ | ---------------------------------------------------------------- | ----------------------------------------------------------------------------------------------------------------- | ------------------------------------------------------------------------------------------------------------- |
> | **4**        | Stable, no noticeable oscillations                               | Converges slowly; at step 100 the median length remains **> 1000 tokens**, and later steps rarely shorten further | Sampling space too narrow; once answers reach a moderate length the policy lacks diversity to push shorter    |
> | **8**        | Moderate, well-controlled fluctuations (see Fig. 6 in the paper) | Good balance: faster length drop than *n = 4* while maintaining accuracy                                          | Trade-off point where learning is both efficient and stable                                                   |
> | **12**       | Large oscillations with a downward accuracy trend                | Very fast: by **step 40** median length falls **< 1000 tokens**; by **step 200** it reaches **≈ 300 tokens**      | Wide sampling space produces aggressive SOL targets, accelerating length shrinkage but destabilising accuracy |
>
> * **Key observation:** Increasing *n* enlarges the candidate pool, so the chosen SOL target becomes more aggressive. This markedly accelerates length reduction but also amplifies accuracy variance.
> * **Practical recommendation:** *n = 8* offers the best balance between rapid convergence and accuracy stability and, critically, fits within the resource budget for larger models such as 7 B parameters.
>
> ---

---

> > ### Comment · Reviewer_2UEk · 2025-08-04
> >
> > Thank you for your response. The additional experiments on the hyperparameters are helpful, and I encourage you to include them in the paper.
> >
> > While I still have some concerns about potential reasoning shortcuts introduced by ShorterBetter, I will retain my original rating to show my support for this work.

---

> > > ### Author Response · Authors · 2025-08-05
> > >
> > > Thank you for your comprehensive review and valuable feedback. We appreciate you highlighting the strengths of our method and the utility of our reasoning trace analysis. We have carefully considered your concerns regarding potential reasoning shortcuts and the choice of hyperparameters and will incorporate the additional experiments and discussions on hyperparameter trade-offs in the future version, which we believe will further clarify our contribution.

---

### Official Review · Reviewer_iAtn · 2025-07-07

**Clarity:** 3
**Significance:** 2
**Originality:** 2
**Rating:** 4
**Confidence:** 4

**Summary:**

ShorterBetter introduces a reward function that explicitly accounts for Optimal Reasoning Length (OL). In addition to answer accuracy, the function assigns a penalty to any generation whose length diverges from the shortest correct solution—or, when no correct solution appears in the roll-outs, from the batch-average length. They assess ShorterBetter on both in-distribution and out-of-distribution mathematics benchmarks and perform ablation studies to isolate and validate the contribution of the length-based penalty.

**Questions:**

Please see weakness.

Also, what do you mean by "may inform the design of more fine-grained and behavior-aware reasoning optimization strategies" (in section 6)?

**Ethical Concerns:**

["NO or VERY MINOR ethics concerns only"]

**Final Justification:**

I maintain my evaluation

**Limitations:**

Yes the authors mentioned future improvements such as 1) applying to larger-scale reasoning model, 2) tasks with non-binary correctness score, 3) more fine-grained reasoning optimization strategies.

**Quality:**

3

**Strengths And Weaknesses:**

### **Strengths:**
- The presentation is clear, and the method is easy to understand.
- The experimental study seems comprehensive, with 4 ID and 4 OOD tasks.
- The authors conduct a comprehensive ablation study with varying values of the hyperparameters in the reward function.

### **Weakness:**
- **Novelty:** ShorterBetter implements an augmented version of GRPO with length regularization on top of the task metrics. Previous works like L1 (Aggarwal et al., 2025) and concurrent works such as ConciseR (Song et al., 2025) have similar notions, where L1 also leverages GRPO to fine-tune LLMs. I saw the authors compare with L1:
```
Existing approaches targeting efficient LLM reasoning primarily focus on explicitly controlling reasoning length, for instance by imposing a length budget based on user specifications (L1; Aggarwal and Welleck, 2025) … In contrast to these explicit control strategies, our work is based on the hypothesis that reasoning models implicitly possess an optimal reasoning length (OL).
```
If I understand correctly, ShorterBetter can also be viewed as a self-supervised version of L1 where the optimal length is defined by the model itself. In this context, the novelty seems insufficient.

- **Robustness:** There are some interesting yet not fully explained conclusions. Specifically, Figure 4 shows that naively adopting the shortest response as the target length leads to training collapse. The resulting accuracies and output lengths are all zeros, even for “Partial.” Can the authors provide more explanation? If the training dynamics are sensitive to the length penalty, how do the authors, in practice, choose the hyperparameters?

### **Minor:**
- I don’t think the section “Output Length after First Appearance of Correct Answer” is necessary. It seems to convey the same information as comparing the total answer length. Also, why “after” and not “before”? Redundancy can also occur before the model reaches the correct answer, no?

---

> ### Author Rebuttal · Authors · 2025-07-31
>
> We thank the reviewer for finding our presentation clear and experimental study concrete. We also appreciate the thoughtful feedbacks regarding the **novelty** and **robustness** of our method. Below are our point-by-point responses:
>
> ---
> ## **On the Novelty of Our Method**
>
> The reviewer has pointed out the previous `L1` (Aggarwal et al., 2025) and concurrent work `ConciseR` (Song et al., 2025), and correctly notes that all three methods operate in the space of RL for efficient reasoning. We also believe that comparing with these two methods allows us to clarify and demonstrate the novelty of `ShorterBetter`.  On a high level:
>
> - `L1` (Aggarwal et al., 2025) focuses on **external controllability**, where a user must specify a length budget.
> - `ConciseR` (Song et al., 2025) uses a **gated, two-stage process** to first improve reasoning and then separately optimize for length.
> - `ShorterBetter`, in contrast, is designed to **autonomously discover an optimal reasoning length**, based on the model's own capabilities and the problem's difficulty, within a single, unified training process.
>
> **Detailed comparison with L1:**
>
> The reviewer suggests `ShorterBetter` can be viewed as a _"self-supervised version of L1."_ While we agree our method is self-supervised, we believe this framing understates the conceptual leap. Notable differences are:
>
> - **Objective:** `L1`'s objective is adherence to a user-defined budget. The model is rewarded for matching a length ($n_{gold}$) provided in the prompt. `ShorterBetter`'s objective is to find the most efficient path to a correct solution -- which can be viewed as self-supervising the model to approximate a (hidden) optimal length. Crucially, unless the user already knows the "most efficient solution" to a problem a priori, the self-supervised target in our approach will also deviate from $n_{gold}$.
> - **Assumption:** In order for the method to work well, `L1` requires the user-defined budget (target) to be informative of the problem at hand, which we argue can be sometimes unrealistic (_how do you set a budget for a math question that you cannot solve?_). On the other hand, `ShorterBetter` is built on the more general hypothesis that an **Optimal Reasoning Length (OL)** exists, which the model learns to find on its own without requiring this prior knowledge.
> - **Prompt dependence:** `L1` requires appending an explicit cue—*“think for n tokens”*—to every training prompt, a design choice that can itself shape the model’s behavior; this interaction is not analyzed in detail in their paper. By contrast,  `ShorterBetter` is completely prompt-agnostic, making it broadly applicable.
>
> **Detailed comparison with ConciseR:**
>
> `ConciseR` is an excellent concurrent work, but it addresses a different problem setting with a different methodology. Notable differences are:
>
> *   **Problem Formulation:** We start with a strong but *overthinking* reasoning model and our goal is to **refine** its reasoning to be more efficient. `ConciseR` starts with a weaker base model and must first **build up** its reasoning capability before optimizing for length. These are fundamentally different starting points and objectives.
> *   **Training Paradigm & Gating:** `ConciseR` uses a **sequential two-stage** framework. Stage 1 improves reasoning, and Stage 2 enforces conciseness. Crucially, its length reward is **conservatively gated**; it only activates when *all* sampled rollouts are correct. In contrast, `ShorterBetter` uses a **single, integrated** process. Our $l_{SOL}$ provides a **flexible learning signal** as long as *at least one* correct response exists, making our method more sample-efficient, especially on challenging problems where 100% correctness per batch is rare.
> *   **Reward Signal:** `ConciseR`'s reward aims for maximum compression (rewarding $L_{Max} - l_i$). `ShorterBetter`'s $l_{SOL}$ reward targets an *optimal point*, penalizing responses that are too long *and* too short.
>
> **To summarize:**
>
> The core novelty of `ShorterBetter` is the introduction of **Sample Optimal Length (SOL)** as a dynamic, self-supervised reward signal. This creates an autonomous and integrated framework that guides a model to discover its own optimal reasoning length, rather than simply obeying an external budget (`L1`) or following a staged compression schedule (`ConciseR`).
>
> ---
> ## **On the Robustness of Our Method**
>
> We thank the reviewer for their insightful question on training robustness and hyperparameter sensitivity. We offer this explanation, supplemented by new experiments, to clarify these critical aspects.
>
> ### **On Training Collapse from Naively Minimizing Length**
>
> The reviewer correctly notes the training collapse shown in our paper (Fig. 4, Left). This collapse is caused by an unstable reward signal. Our ablation studies confirm that naively targeting the shortest length is not viable:
>
> 1.  **No Correctness Reward, Shortest Length Target:** As shown in our paper, if the reward only minimizes length, the model quickly learns to produce empty outputs, causing accuracy to drop to zero.
>
> 2.  **"Partial" Shortest Length Target:** Using the shortest length as a target only when no correct answers are found also fails (even with a reward term present). This creates a feedback loop where the model sacrifices correctness for length, leading to a training collapse.
>
> 3.  **Correctness Reward + Always Shortest Length Target:** In a new ablation, we kept the correctness reward ($\alpha=2$) but always set the length target to the shortest response in a group. This also crashed the training. Our logs show accuracy and length rapidly declining to zero. Even with a correctness reward, a single short (but incorrect) response heavily penalizes all longer (and potentially correct) responses, creating a fatal bias toward underthinking.
>
> These experiments show that the length target design is critical. Our proposed **SOL**, anchored to the shortest *correct* response and defaulting to the *average* length in failure cases, provides the necessary stability to avoid these collapse scenarios.
>
> ---
>
> ### **On the Practical Choice of Hyperparameters**
>
> The hyperparameter $\alpha$ controls the trade-off between accuracy and efficiency. Its value relative to the length penalty $\beta$ (kept at 0.001) is the key factor. We tested new $\alpha$ values to illustrate its impact.
>
> **Performance of `ShorterBetter-1.5B` with Varying $\alpha$**
>
> | Dataset  | $\alpha$=0.1 Acc | $\alpha$=0.1 Len | $\alpha$=2.0 Acc | $\alpha$=2.0 Len | $\alpha$=5.0 Acc | $\alpha$=5.0 Len |
> | :--- | :---: | :---: | :---: | :---: | :---: | :---: |
> | AMC      | 0.43         | 2503      | 0.57         | **1946** | **0.58** | 3077      |
> | Minerva  | 0.20         | 1785      | **0.28** | **1147** | 0.24         | 2249      |
> | Olympiad | 0.30         | 2903      | **0.38** | **1814** | 0.35         | 3919      |
> | AIME     | 0.13         | 5395      | 0.20         | **2703** | **0.23** | 5274      |
>
>
> We present new results for $\alpha=0.1$ and $\alpha=5.0$ to complement the results for $\alpha \in \{0, 1, 2\}$ in the main paper.
>
> For alpha = 0.1, the training log shows that accuracy follows a declining trend over 200 training steps. During this time, the median output length decreases rapidly.
> For alpha = 5.0, the training log indicates a clear upward trend in accuracy throughout the training process. The median output length also trends downward, decreasing from 5000 to below 2000 tokens.
>
> Our findings provide an intuition for selecting $\alpha$:
>
> * **A high $\alpha$ (e.g., 5.0)** places a strong emphasis on correctness. As seen in our training dynamics log and the table above, this leads to stable accuracy gains but a weaker length-reduction, resulting in longer outputs compared to the more balanced setting. The accuracy trend is clearly positive during training.
> * **A low $\alpha$ (e.g., 0.1)** makes the length penalty term negatively influential. This creates a risk of degrading the model's reasoning capability, as the reward for correctness may not be sufficient to maintain performance. The training dynamics log shows a downward trend in accuracy. This instability can lead to worse length reduction, as seen in the table, because the model fails to learn effective, concise reasoning paths.
> * **A balanced $\alpha$ (e.g., 1.0 or 2.0)** provides a strong enough correctness signal to preserve or improve accuracy while still exerting significant pressure to reduce output length. This achieves the best overall trade-off demonstrated in our paper.
>
> In practice, the optimal choice depends on the specific goal (e.g., prioritizing efficiency vs. accuracy) and specific model family. Based on our results, we recommend practitioners start with $\alpha$ in the range of [2, 5] and tune as needed.
>
> ---
> ### **On the "Length After First Appearance" Metric**
>
> While we agree that our “Length after First Appearance” metric is correlated with total length,, we believe it provides a unique and valuable perspective (as pointed out by Reviewer 2UEK) on a **specific pattern of overthinking**: a model's inability to terminate its reasoning process after it has already found a correct solution.
>
> The reviewer correctly notes that redundancy can occur both before the answer is found ("pre-solution") and after ("post-solution"). Our metrics are designed to disentangle these two effects. Total length captures the sum of both inefficiencies. In contrast, our "Length After" metric focuses on the post-solution component. Therefore, we believe that the total length and the "Length After" metrics offer complementary views. We will revise Section 5.2 to make this clear.
>
> ---
> ### **Clarification on Sec 6**
>
> Finally, our statement at the end of Sec 6 means that we encourage future works in efficient reasoning to also adopt similar reasoning trace analysis, which provides more fine-grained and detailed feedbacks to understand the model behavior than simply looking at overall length reduction.

---

### Note · Authors · 2025-08-15

We sincerely thank all reviewers for their feedback. In our rebuttal, we have provided additional experiments and detailed explanations to clarify and address the following points.

- **Hyperparameter Robustness:** Building on our initial analysis, we provided new experiments in response to feedback from Reviewers iAtn, 2UEk, and 3dRz. These results illustrate the accuracy-efficiency trade-off controlled by the hyperparameter $\alpha$ and demonstrate that a moderate rollout size (n=8) provides an optimal balance between rapid convergence and training stability.

- **Baseline Comparisons and Ablations:** To provide a clear comparison requested by Reviewers 5QGT and iAtn, we conducted a new experiment against the "Training Efficient" baseline. By integrating its reward function into our framework, we isolated the contribution of our SOL design, showing it significantly outperforms the baseline. Furthermore, we deepened our ablation of the reward mechanism, confirming that alternative designs lacking our correctness-based anchor are unstable and lead to training collapse.

- **LLM-as-a-Judge Validation:** Reviewer 3dRz questioned the accuracy of the LLM judge used in our reasoning trace analysis. To address this, we conducted a human-annotation study and found a high inter-annotator agreement of 94.23% at the sentence level, which validates our LLM judge and strengthens our conclusions about the structural improvements of our method.

- **Clarifying novelty:** We provided detailed comparisons between our paper and two related works (L1 and ConciseR) brought up by reviewer iAtn. We emphasized crucial differences in our assumptions, as well as our paper’s novelty in reward design.

**Planned revisions**

In the camera-ready version, we will:

- Expand the related works section to better position our contributions within concurrent efficient reasoning research.
- Include hyperparameter sweeps and ablation results in the appendix.
- Include the human verification results collected during the rebuttal and provide additional case studies of the reasoning traces in the appendix.

We believe that these revisions enhance the strength of our paper, and we are grateful to the reviewers for their feedback. We hope that our novel SOL design and reasoning trace analysis framework will contribute to the field of efficient reasoning and encourage further research and discussion.

---

### Decision · Program_Chairs · 2025-09-17

**Decision:**

Accept (poster)

**Comment:**

The paper proposes controlling response lengths in CoT reasoning for avoiding overthinking. This is done via RL with length regularization.
Reviewers note clear presentation, comprehensive experiments, empirical strength, good ablations. On the other hand, more ablations further evaluation also on different models and many hyper parameters. Also concurrent similar work is mentioned.
Given the good experimental results and reviewer consensus, the paper merits acceptance at NeurIPS.